# All Jordanian deformations of
# the $AdS_5 \times S^5$ superstring

Riccardo Borsato,   Sibylle Driezen

*Instituto Galego de Física de Altas Enerxías (IGFAE) and Departamento de Física de Partículas,*

*Universidade de Santiago de Compostela*

`riccardo.borsato@usc.es, sib.driezen@gmail.com`

**Abstract**

We explicitly construct and classify all Jordanian solutions of the classical Yang-Baxter equation on $\mathfrak{psu}(2,2|4)$, corresponding to Jordanian Yang-Baxter deformations of the $AdS_5 \times S^5$ superstring. Such deformations preserve the classical integrability of the underlying sigma-model and thus are a subclass of all possible integrable deformations. The deformations that we consider are divided into two families, unimodular and non-unimodular ones. The former ensure that the deformed backgrounds are still solutions of the type IIB supergravity equations. For the simplest unimodular solutions, we find that the corresponding backgrounds preserve a number $N < 32$ of supercharges that can be $N = 12, 8, 6, 4, 0$.

# 1 Introduction

Recent years have seen an upsurge in the understanding of integrable deformations of two-dimensional sigma-models, including their possible classifications and applications. Well-known cases are TsT-transformations [1–3], Yang-Baxter deformations [4–9], and $\lambda$-deformations [10–12]. Major motivations for their study follow from developing a fundamental understanding of integrable two-dimensional field theories, their relation with generalised worldsheet dualities, and their prospect in generalising the AdS/CFT correspondence to non-maximally supersymmetric cases whilst preserving the computational power of integrability (see e.g. the reviews [13, 14]). In particular, when the sigma-model describes the dynamics of a string then an integrable deformation on its worldsheet will deform the target-space background destroying many of its (super)isometries. Whether or not the deformed background will still solve the supergravity equations of motion is now well-understood for large classes of deformations (see e.g. [15–20]). This thus opens exciting possibilities for applications in particular to the canonical example of

the $AdS_5 \times S^5$ string, with hopes to find deformations of $\mathcal{N} = 4$ super-Yang-Mills that remain integrable in the planar limit [21].

In this paper, we will focus on a particular type of deformations, called Jordanian, which belong to the homogeneous deformations of the Yang-Baxter class [8,9]. Homogeneous deformations are generated by a linear operator $R$ that acts on the Lie (super)algebra of (super)isometries $\mathfrak{g}$ of an integrable sigma-model, and they preserve integrability if $R$ is antisymmetric as in eq. (A.2) of appendix A and satisfies the classical Yang-Baxter equation (CYBE) (see eq. (A.3) of appendix A).[1] The CYBE admits a rich number of antisymmetric solutions, all of which will lead to different deformed target-space backgrounds of the string sigma-model. These backgrounds will be supergravity solutions when the $R$-matrix is unimodular [15, 22, 23], which is a simple linear constraint on $R$, see eq. (A.7). When instead $R$ is non-unimodular, they will solve the modified (or generalised) supergravity equations identified in [24, 25]. Within the homogeneous Yang-Baxter models, $R$-matrices of Jordanian type are built by identifying a bosonic subalgebra of $\mathfrak{g}$ constructed from a Cartan element $\mathsf{h}$ and a root $\mathsf{e}$ satisfying $[\mathsf{h}, \mathsf{e}] = \mathsf{e}$. When $\mathsf{h}$ and $\mathsf{e}$ are the only elements in its construction, the Jordanian $R$-matrix is of rank-2 and will in fact be non-unimodular. However, it was found in [26] that at least some of these rank-2 cases can be extended to unimodular $R$-matrices by employing, next to $\mathsf{h}$ and $\mathsf{e}$, also fermionic generators (supercharges) of the superalgebra $\mathfrak{g}$ in its construction.

Our focus on the homogeneous deformations of Jordanian type originates from several motivations. In general, homogeneous deformations generalise the well-known TsT (T-duality-shift-T-duality) transformations [1–3] to the case where the subalgebra participating in the construction of $R$ is non-abelian [27]. For TsT, this subalgebra is abelian and corresponds to the commuting isometries along which the T-dualities are performed. An important property of TsT is that the corresponding deformed models with periodic worldsheet boundary conditions can be reformulated as undeformed models with twisted worldsheet boundary conditions [2,3,28] that are local. In the case of "diagonal" TsT-models, where the object causing the twisting is diagonal, this has been crucial in the understanding of their spectral problem on both sides of deformed $AdS_5$/SYM by means of integrability methods [29–31]. Recently, the reformulation in terms of a (local) twisted model has been achieved for generic homogeneous Yang-Baxter deformations [32].[2] In contrast to other options, it was found that the Jordanian $R$-matrices lead to a twist that is always diagonalisable, and therefore one could hope to apply similar integrability techniques that worked for TsT to tackle their spectral problem. This has been done successfully at the semi-classical level in [36] for a specific Jordanian deformation of $AdS_5 \times S^5$ (which corresponds to our $R_1$ below with $a = 0, b = -1/2$ and to $\bar{R}_1$ with $a = 0, b = -1/2, q_{2,22}^+ = 0$ for its unimodular version).

In this paper, we will continue the study of string sigma-models deforming the canonical $AdS_5 \times S^5$ background[3] and therefore take $\mathfrak{g} = \mathfrak{psu}(2, 2|4)$. To diversify the applications of Jordanian models, we will classify all antisymmetric solutions to the CYBE that are of Jordanian type. This includes a classification of all canonical rank-2 Jordanian $R$-matrices as well as their bosonic extensions. We will find that they are at most of rank-6. For all of the bosonic $R$-matrices, we will explicitly construct only those fermionic extensions that ensure unimodularity and thus a corresponding supergravity background that is well-behaved. We will pay particular

---

[1] For the inhomogeneous version, the antisymmetric $R$ must instead solve the modified Yang-Baxter equation [4–7].

[2] For earlier work giving a reformulation in terms of undeformed models with non-local twisted boundary conditions see [33–35].

[3] Although our main interest is in $AdS_5 \times S^5$, some of our results are useful to classify also deformations of backgrounds $AdS_n \times M$ with $n < 5$. In fact some of our $R$-matrices are constructed with generators of $\mathfrak{so}(2, 4)$ that are also elements of the isometry groups of $AdS_n, n < 5$, and therefore they can be used to generate corresponding deformations.

attention to simplifying our results as much as possible, including the identification of equivalent solutions, by means of inner automorphisms of $\mathfrak{g} = \mathfrak{psu}(2, 2|4)$.

The paper is organised as follows. We present a summary of our main results in section 2. In section 3, we define Jordanian $R$-matrices including their bosonic and fermionic extensions, as well as the unimodularity condition. This in particular identifies the necessary requirements that the subalgebra used to construct the $R$-matrix has to satisfy. In section 4, we present our classification of the bosonic rank-2 $R$-matrices. These results are summarised in Table 1 and 2. In section 5, we will construct their bosonic higher-rank extensions—which only exist for certain rank-2 cases—summarised in the rest of section 2. In section 6, we then construct all the fermionic extensions that ensure unimodularity. Again in many of the rank-2 cases, we will see that they actually do not admit a unimodular extension, while all higher-rank cases do. In section 7, we revive our string theory motivation, and identify for all possible unimodular extensions of the rank-2 $R$-matrices the number of (super)isometries that are preserved in the corresponding deformed supergravity background. The number of preserved superisometries is summarised in Table 3, while the inclusion of bosonic isometries is presented in Table 6. We end with some conclusions in section 8. Appendix A collects facts and conventions on homogeneous $R$-matrices, including the corresponding deformed sigma-model, while appendix B collects our conventions on the $\mathfrak{psu}(2, 2|4)$ algebra and presents the explicit matrix realisation that we used.

## 2    Summary of the results

For the reader's convenience, in this section we present a summary of our main results. All our results are presented modulo inner automorphisms of the algebra, or modulo transformations that leave the $R$-matrix invariant. Inner automorphisms of the algebra map equivalent solutions of the $R$-matrices to each other: from the point of view of the sigma model the two deformations would be related by field redefinitions, and from the point of view of the 10-dimensional background by coordinate transformations.

The bosonic Jordanian $R$-matrices may be grouped by their rank. We find that we can have bosonic Jordanian solutions of rank 2, 4 and 6. In the following we present the results for the bosonic $R$-matrices and their unimodular extensions.

### Rank-2

The simplest possible case is that of rank 2, with $R$-matrices of the form $r = \mathsf{h} \wedge \mathsf{e}$. Here we are using a notation to write the $R$-matrix that is reviewed in appendix A. In Tables 1 and 2 we collect all possible rank-2 $R$-matrices of $\mathfrak{so}(2, 4)$. In Table 1 we list the ones that admit a unimodular extension, while in Table 2 the ones that do not admit it. It turns out that only $R$-matrices with $\mathsf{e} = p_0 + p_3$ or $\mathsf{e} = p_0$ admit a unimodular extension.

Because we are interested in all possible deformations of $AdS_5 \times S^5$, in principle we can always shift $\mathsf{h}$ and $\mathsf{e}$ by elements of the Lie algebra of the isometry group of $S^5$. In other words, given $r = \mathsf{h} \wedge \mathsf{e}$ we can always construct $r' = \mathsf{h}' \wedge \mathsf{e}'$ where $\mathsf{h}' = \mathsf{h} + \mathsf{t}_1, \mathsf{e}' = \mathsf{e} + \mathsf{t}_2$ and $\mathsf{t}_i \in \mathfrak{so}(6) \subset \mathfrak{psu}(2, 2|4)$ with $[\mathsf{t}_1, \mathsf{t}_2] = 0$. This option has the interpretation of applying first a sequence of 3 TsT transformations along $(\mathsf{t}_1, \mathsf{t}_2), (\mathsf{h}, \mathsf{t}_2), (\mathsf{t}_1, \mathsf{e})$ (the relative order among the 3 TsT's is inconsequential) followed by the Jordanian deformation $r = \mathsf{h} \wedge \mathsf{e}$.[4] At the level of bosonic $R$-matrices, also for higher rank we always have the option of shifting the $\mathfrak{so}(2, 4)$

---

[4]See e.g. section 3.1 of [37].

| $R$ | h | e | residual $\mathrm{Inn}(\mathfrak{so}(2,4))$ |
|---|---|---|---|
| 1 | $(1+b)D + bJ_{03} + aJ_{12}$ | $p_0 + p_3$ | $J_{12},\, D+J_{03}$ (and $p_1, p_2$ if $a=0$ and $b=-1$; and $J_{01}-J_{13},\, J_{02}-J_{23}$ if $a=b=0$; and $p_0-p_3,\, k_0+k_3$ if $b=-1/2$) |
| 2 | $\frac{1}{2}(D-J_{03}) + aJ_{12} + \alpha(p_0-p_3)$ | $p_0 + p_3$ | $J_{12},\, p_0 - p_3$ |
| 3 | $-J_{03} + \alpha p_1$ | $p_0 + p_3$ | $p_1, p_2$ |
| 4 | $D + \alpha(J_{01}-J_{13})$ | $p_0 + p_3$ | $J_{01}-J_{13},\, J_{02}-J_{23}$ |
| 5 | $\frac{1}{2}(D-J_{03}) + aJ_{12} + b(k_0+k_3+2p_3)$ | $p_0 + p_3$ | $J_{12},\, k_0+k_3-p_0+p_3$ |
| 6 | $D + aJ_{12}$ | $p_0$ | $J_{12}$ (and $J_{13}, J_{23}$ if $a=0$) |

Table 1: Rank-2 bosonic Jordanian deformations of the form $r = \mathsf{h} \wedge \mathsf{e}$ that admit a unimodular extension. The convention is that the parameter $\alpha$ squares to 1 ($\alpha^2 = 1$) and $a, b \in \mathbb{R}$ are free. In the last column we write the residual inner automorphisms in $\mathfrak{so}(2,4)$ that (together with $\mathfrak{so}(6)$) leave h and e invariant. In those lists we omit e itself, which always corresponds to an isometry.

| h | e |
|---|---|
| $D + aJ_{03}$ | $p_1$ |
| $D + \alpha(J_{02}-J_{23})$ | $p_1$ |
| $D + \alpha(p_0+\beta p_3) + \beta J_{03}$ | $p_1$ |
| $-J_{03} + aD$ | $J_{02}-J_{23}$ |
| $-J_{03} + \alpha p_1$ | $J_{02}-J_{23}$ |
| $-J_{03} - D + \alpha(p_0+p_3)$ | $J_{02}-J_{23}$ |
| $D - J_{03}$ | $ap_1 + bp_2 + J_{01} - J_{13}$ |
| $D + aJ_{12}$ | $p_3$ |
| $2D - J_{03}$ | $p_0 - p_3 + J_{01} - J_{13}$ |
| $D - J_{03} - 2a\alpha J_{12} + a(k_0+k_3+2p_3)$ | $\alpha p_2 + J_{01} - J_{13}$ |

Table 2: Rank-2 bosonic Jordanian deformations of the form $r = \mathsf{h} \wedge \mathsf{e}$ that do *not* admit a unimodular extension. The convention is that the parameters $\alpha, \beta$ square to 1, ($\alpha^2 = \beta^2 = 1$), and $a, b \in \mathbb{R}$ are free.

elements by elements of $\mathfrak{so}(6)$, so we will not repeat this discussion. Importantly, the unimodular extensions will be affected by these possible $\mathfrak{so}(6)$ shifts, but we will not analyse this further.

Going back to the rank-2 solutions of Table 1 that admit a unimodular extension, in the last column we list the inner automorphisms of $\mathfrak{so}(2,4)$ that are residual isometries under the deformation and that leave h and e invariant. It turns out that this information is useful also to simplify as much as possible the unimodular extensions of these rank-2 solutions. In particular, a unimodular $R$-matrix will be of the form

$$r = \mathsf{h} \wedge \mathsf{e} - \frac{i}{2}(\mathsf{Q}_1 \wedge \mathsf{Q}_1 + \mathsf{Q}_2 \wedge \mathsf{Q}_2), \tag{2.1}$$

where $\mathsf{Q}_{\mathsf{i}}$, $\mathsf{i} = 1, 2$ are odd elements of $\mathfrak{psu}(2,2|4)$ of the form

$$\mathsf{Q}_{\mathsf{i}} = q^+_{\mathsf{i},\alpha a} \mathcal{Q}^{\alpha a}_+ + i\, q^-_{\mathsf{i},\alpha a} \mathcal{Q}^{\alpha a}_-, \tag{2.2}$$

where $\mathcal{Q}^{\alpha a}_\pm \equiv \overline{Q}^{\alpha a} \pm \epsilon^{\alpha\beta} Q_{\beta a}$ and $q^\pm_{\mathsf{i},\alpha a}$ are real numbers. We refer to appendix B for our conventions on $\mathfrak{psu}(2,2|4)$.

It turns out that the unimodular extensions of rank-2 solutions come in three types. First we have $\bar{R}_1, \bar{R}_2, \bar{R}_3, \bar{R}_4, \bar{R}_5$ as unimodular extensions of the corresponding bosonic $R$-matrices

$R_1, R_2, R_3, R_4, R_5$. This type of extension works only if there is no $J_{12}$ in $\mathsf{h}$, so that one has to set $a = 0$ in $R_1, R_2, R_5$. We find that we can set to zero all coefficients $q^\pm_{i,\alpha a}$ except three of them that satisfy

$$q^+_{1,21} = \frac{1}{\sqrt{2}}, \qquad (q^+_{2,22})^2 + (q^-_{2,21})^2 = \frac{1}{2} \ . \tag{2.3}$$

Notice that this solution admits one continuous parameter. It is a physical parameter, in fact when $q^+_{2,22} = 0$ there is an enhancement of the number of superisometries.

We then have the second type of extension, that can be constructed when there is a non-trivial contribution of $J_{12}$ in $\mathsf{h}$. That means that we assume $a \neq 0$ in $R_1, R_2, R_5$ and construct the unimodular $R$-matrices $\bar{R}_{1'}, \bar{R}_{2'}, \bar{R}_{5'}$. In this case all coefficients $q^\pm_{i,\alpha a}$ can be set to zero except two of them

$$q^-_{2,21} = q^+_{1,21} = \frac{1}{\sqrt{2}} \ . \tag{2.4}$$

Taking the $a \to 0$ limit one recovers the previous type of extension with the extra condition $q^+_{2,22} = 0$.

Finally, $R_6$ with $\mathsf{e} = p_0$ stands out on its own. It admits the last type of unimodular extension $\bar{R}_6$ where all coefficients are 0 except

$$q^-_{1,11} = q^+_{2,11} = q^-_{1,22} = -q^+_{2,22} = \frac{1}{2} \ . \tag{2.5}$$

In Table 3 we summarise the number of superisometries that are preserved in the deformation for each of the unimodular extensions of the rank-2 $R$-matrices. A table that includes also the residual bosonic isometries is given in the main text, see Table 6.

| $\bar{R}$ | conditions | supercharges |
|---|---|---|
| | $a = 0$ | 0 |
| 1 | $a = 0, b = -1/2$ | 8 |
| | $a = 0, b = -1/2, q^+_{2,22} = 0$ | 12 |
| 1' | $a \neq 0$ | 0 |
| 2 | $a = 0$ | 4 |
| | $a = 0, q^+_{2,22} = 0$ | 6 |
| 2' | $a \neq 0$ | 0 |
| 3 | $-$ | 0 |
| 4 | $-$ | 0 |
| 5 | $a = 0$ | 0 |
| 5' | $a \neq 0$ | 0 |
| 6 | $-$ | 0 |

Table 3: The number of independent supercharges $T_{\bar{A}} \in \mathfrak{psu}(2,2|4)$ that satisfy $\mathrm{ad}_{T_{\bar{A}}} R = R \, \mathrm{ad}_{T_{\bar{A}}}$ for the unimodular extended rank-2 Jordanian $R$-matrices of the form $r = \mathsf{h} \wedge \mathsf{e} - \frac{i}{2}(\mathsf{Q}_1 \wedge \mathsf{Q}_1 + \mathsf{Q}_2 \wedge \mathsf{Q}_2)$. Such elements represent superisometries of the deformed supergravity background. If a parameter is not specified, it is assumed to be generic (modulo constraints such as (6.21)).

## Rank-4

Higher-rank bosonic Jordanian $R$-matrices can be constructed only in the case of $\mathsf{e} = p_0 + p_3$. Rank-4 solutions are of the form $r = \mathsf{h} \wedge \mathsf{e} + \mathsf{e}_+ \wedge \mathsf{e}_-$, they are all listed in Table 4 and they all admit a unimodular extension.

| $R$ | h | $e_+$ | $e_-$ |
|---|---|---|---|
| 7 | $(1+b)D + b\,J_{03}$ | $a\,p_1 + p_2$ | $j_{023}$ |
| 8 | $\frac{2}{3}D - \frac{1}{3}J_{03}$ | $a\,p_1 + p_2$ | $b\,(p_0 - p_3) + j_{023}$ |
| 9 | $\frac{1}{2}(D - J_{03})$ | $p_1 + b_+ p_2 + d_+ j_{023}$ | $b_- p_2 + d_- j_{023}$ <br> $+(1 - b_+ d_- + b_- d_+)j_{013}$ |
| 10 | $-J_{03} + \alpha\,p_1$ | $-j_{023}$ | $a\,p_1 + p_2$ |
| 11 | $D + \alpha j_{013}$ | $p_2$ | $a j_{013} + j_{023}$ |
| 12 | $\frac{1}{2}(D - J_{03}) + aJ_{12}$ <br> $+\frac{\alpha a}{2}(k_0 + k_3 + 2p_3)$ | $p_1 + \alpha\,j_{023}$ | $-\frac{1}{2}(\alpha p_2 - j_{013})$ |

Table 4: Rank-4 bosonic Jordanian deformations of the form $r = \mathsf{h} \wedge \mathsf{e} + \mathsf{e}_+ \wedge \mathsf{e}_-$. In all these cases we have $\mathsf{e} = p_0 + p_3$. To save space we are using the shorthand notation $j_{\mu\nu\rho} = J_{\mu\nu} - J_{\nu\rho}$. All these $R$-matrices admit unimodular extensions, with no further restriction of the above parameters. When we have the parameter $\alpha$, it is assumed that $\alpha^2 = 1$, while all other parameters with latin letters are free real numbers.

The unimodular extensions of these $R$-matrices are constructed with 4 (rather than just 2) odd elements of $\mathfrak{psu}(2, 2|4)$

$$r = \mathsf{h} \wedge \mathsf{e} + \mathsf{e}_+ \wedge \mathsf{e}_- - \frac{i}{2} \sum_{\mathsf{l}=1,2} (\mathsf{Q}_1^\mathsf{l} \wedge \mathsf{Q}_1^\mathsf{l} + \mathsf{Q}_2^\mathsf{l} \wedge \mathsf{Q}_2^\mathsf{l}), \tag{2.6}$$

where we added a label $\mathsf{l} = 1, 2$ and we still take

$$\mathsf{Q}_i^\mathsf{l} = q_{i,\alpha a}^{+,\mathsf{l}} \mathcal{Q}_+^{\alpha a} + i\, q_{i,\alpha a}^{-,\mathsf{l}} \mathcal{Q}_-^{\alpha a}. \tag{2.7}$$

The unimodular extensions of $R_7, R_8, R_9, R_{10}, R_{11}$ are of the same type and can be taken as follows. The coefficients for $\mathsf{Q}_1^1$ and $\mathsf{Q}_2^1$ can be simplified as in the extension $\bar{R}_1$, meaning that we can set

$$q_{1,21}^{+,1} = \frac{1}{\sqrt{2}}, \qquad (q_{2,22}^{+,1})^2 + (q_{2,21}^{-,1})^2 = \frac{1}{2}\,. \tag{2.8}$$

For $\mathsf{Q}_1^2$ and $\mathsf{Q}_2^2$ we can have in principle 7 non-trivial parameters turned on, satisfying the conditions (6.14) and with the extra choice of $q_{1,24}^{\pm,2} = q_{1,23}^{-,2} = q_{i,21}^{+,2} = 0$.

The result for the unimodular extension of $R_{12}$ is simpler to present, because all coefficients can be set to zero except

$$q_{1,21}^{-,1} = -q_{2,21}^{+,1} = q_{1,22}^{-,2} = -q_{2,22}^{+,2} = \frac{1}{\sqrt{2}}\,. \tag{2.9}$$

Notice that there is no continuous parameter left in the odd elements used to construct this $R$-matrix.

## Rank-6

There are two possible types of rank-6 bosonic solutions. They are of the form $r = \mathsf{h} \wedge \mathsf{e} + \mathsf{e}_{+1} \wedge \mathsf{e}_{-1} + \mathsf{e}_{+2} \wedge \mathsf{e}_{-2}$ and in both cases

$$\mathsf{h} = (1+b)D + bJ_{03}, \qquad \mathsf{e} = p_0 + p_3. \tag{2.10}$$

The first option is to take

$$R_{13}: \quad \mathsf{e}_{+1} = ap_1 + p_2, \qquad \mathsf{e}_{-1} = J_{02} - J_{23}, \qquad \mathsf{e}_{+2} = p_1, \qquad \mathsf{e}_{-2} = J_{01} - J_{13} - a(J_{02} - J_{23}). \tag{2.11}$$

The second option is to set

$$R_{14}: \quad b = -\frac{1}{2}, \qquad \mathsf{e}_{\pm\mathsf{a}} = a_{\pm\mathsf{a}}p_1 + b_{\pm\mathsf{a}}p_2 + c_{\pm\mathsf{a}}\,j_{013} + d_{\pm\mathsf{a}}\,j_{023}, \tag{2.12}$$

while imposing the conditions

$$\begin{aligned}
a_{\pm 1}c_{\pm 2} + b_{\pm 1}d_{\pm 2} - b_{\pm 2}d_{\pm 1} - a_{\pm 2}c_{\pm 1} &= 0, \\
a_{\pm 1}c_{\mp 2} + b_{\pm 1}d_{\mp 2} - b_{\mp 2}d_{\pm 1} - a_{\mp 2}c_{\pm 1} &= 0, \\
a_{+\mathsf{a}}c_{-\mathsf{a}} + b_{+\mathsf{a}}d_{-\mathsf{a}} - a_{-\mathsf{a}}c_{+\mathsf{a}} - b_{-\mathsf{a}}d_{+\mathsf{a}} &= 1, \quad \mathsf{a} = 1, 2.
\end{aligned} \tag{2.13}$$

The unimodular extensions of rank-6 solutions are constructed with 6 odd elements of the superalgebra

$$r = \mathsf{h} \wedge \mathsf{e} + \mathsf{e}_{+1} \wedge \mathsf{e}_{-1} + \mathsf{e}_{+2} \wedge \mathsf{e}_{-2} - \frac{i}{2}\sum_{l=1}^{3}(\mathsf{Q}_1^l \wedge \mathsf{Q}_1^l + \mathsf{Q}_2^l \wedge \mathsf{Q}_2^l). \tag{2.14}$$

The coefficients must again solve the constraints given in (6.14). For $\mathsf{Q}_i^1$ we may again simplify the solution to (2.8), and for $\mathsf{Q}_i^2$ we can set $q_{1,24}^{\pm,2} = q_{1,23}^{-,2} = q_{i,21}^{+,2} = 0$. We were not able to simplify the other coefficients further.

# 3  Extended Jordanian $R$-matrices

Given a Lie superalgebra $\mathfrak{g}$, we identify a Cartan element $\mathsf{h}$ and the generator $\mathsf{e}$ of a positive root. Both of them will be of even grading (i.e. $\deg(\mathsf{h}) = \deg(\mathsf{e}) = 0$), in other words they belong to a standard Lie algebra. The assumption is that $\mathsf{h}$ and $\mathsf{e}$ span a subalgebra of $\mathfrak{g}$ with

$$[\mathsf{h}, \mathsf{e}] = \mathsf{e}. \tag{3.1}$$

When this is the case, they identify a so-called "Jordanian solution" of the classical Yang-Baxter equation that is given by

$$r = \mathsf{h} \wedge \mathsf{e}. \tag{3.2}$$

We refer to appendix A for our conventions on $R$-matrices.

Given a Jordanian $R$-matrix as above, it is possible to construct "extended Jordanian" solutions following Tolstoy [38]. The extra ingredients are $N$ pairs of generators in $\mathfrak{g}$ that we will denote as $\{\mathsf{e}_i, \mathsf{e}_{-i}\}$ with $i = 1, \ldots, N$, where $\mathsf{e}_i$ (resp. $\mathsf{e}_{-i}$) corresponds to a positive (resp. negative) root. These extra generators may be of even or odd grading, but they must satisfy $\deg(\mathsf{e}_i) = \deg(\mathsf{e}_{-i})$. Moreover, the following graded commutation relations must hold[5]

$$[\mathsf{e}_{\pm i}, \mathsf{e}] = 0, \qquad [\mathsf{h}, \mathsf{e}_{\pm i}] = (\tfrac{1}{2} \pm \xi_i)\mathsf{e}_{\pm i}, \qquad [[\mathsf{e}_k, \mathsf{e}_l]] = \delta_{k,-l}\,\mathsf{e}, \tag{3.3}$$

with $k > l \in \{\pm 1, \pm 2, \ldots, \pm N\}$ and $\xi_i \in \mathbb{C}$. Here $[[,]]$ denotes the graded commutator, see appendix A. The $N$-extended Jordanian $R$-matrix is then constructed as

$$r = \mathsf{h} \wedge \mathsf{e} - \sum_{i=1}^{N} \mathsf{e}_{-i} \wedge \mathsf{e}_i, \tag{3.4}$$

where we used the graded wedge product[6]

$$\mathsf{a} \wedge \mathsf{b} = \mathsf{a} \otimes \mathsf{b} - (-1)^{\deg(\mathsf{a})*\deg(\mathsf{b})}\,\mathsf{b} \otimes \mathsf{a}. \tag{3.5}$$

---

[5] Compared to [38] here we use the parameter $\xi$ which is related to the parameter $t$ as $t = \frac{1}{2} - \xi$.

[6] The definition implies that if $\mathsf{a}, \mathsf{b}$ are even then $\mathsf{a} \wedge \mathsf{b} = -\mathsf{b} \wedge \mathsf{a}$ while if they are odd they $\mathsf{a} \wedge \mathsf{b} = +\mathsf{b} \wedge \mathsf{a}$.

See eq. (A.4) to map $r$ to a matrix $R^{IJ}$, with $I, J = 1, \ldots, \dim\mathfrak{g}$.

As already noted in [15], using the above graded commutation relations, one can straightforwardly check the unimodularity condition (A.7) (which, we recall, gives rise to supergravity backgrounds) for a generic $N$-extended Jordanian $R$-matrix, obtaining

$$R^{IJ}[[T_I, T_J]] = 2(-1 - N_0 + N_1)\mathsf{e}, \tag{3.6}$$

where $N_0, N_1$ are respectively the numbers of even and odd extra pairs of generators, so that $N_0 + N_1 = N$. Unimodularity (A.7) then implies $N_1 = N_0 + 1$. Starting from a Jordanian $R$-matrix of the form $r = \mathsf{h} \wedge \mathsf{e}$, a minimal extension that makes it unimodular will therefore be

$$r = \mathsf{h} \wedge \mathsf{e} - \mathsf{e}_+ \wedge \mathsf{e}_-, \tag{3.7}$$

with both $\mathsf{e}_\pm$ of odd grading (i.e. $\deg(\mathsf{e}_\pm) = 1$, so that $N_0 = 0, N_1 = 1$) and

$$[\mathsf{e}_\pm, \mathsf{e}] = 0, \qquad [\mathsf{h}, \mathsf{e}_\pm] = (\tfrac{1}{2} \pm \xi)\mathsf{e}_\pm, \qquad \{\mathsf{e}_+, \mathsf{e}_-\} = \mathsf{e}, \tag{3.8}$$

with $\xi \in \mathbb{C}$. In view of later calculations, we find it convenient to redefine these odd elements as

$$\mathsf{Q}_1 = \frac{1}{\sqrt{2}}(\mathsf{e}_+ - i\,\mathsf{e}_-), \qquad \mathsf{Q}_2 = \frac{1}{\sqrt{2}}(i\,\mathsf{e}_+ - \mathsf{e}_-), \tag{3.9}$$

so that the (anti)commutation relations become

$$[\mathsf{Q}_\mathsf{i}, \mathsf{e}] = 0, \qquad [\mathsf{h}, \mathsf{Q}_\mathsf{i}] = \tfrac{1}{2}\mathsf{Q}_\mathsf{i} - \varepsilon_{\mathsf{ij}}\check{\xi}\,\mathsf{Q}_\mathsf{j}, \qquad \{\mathsf{Q}_\mathsf{i}, \mathsf{Q}_\mathsf{j}\} = -i\delta_{\mathsf{ij}}\mathsf{e}, \tag{3.10}$$

with $\mathsf{i}, \mathsf{j} = 1, 2$, $\check{\xi} = i\xi$ and the antisymmetric tensor $\varepsilon_{12} = -\varepsilon_{21} = 1$. In this basis the $R$-matrix reads

$$r = \mathsf{h} \wedge \mathsf{e} - \frac{i}{2}(\mathsf{Q}_1 \wedge \mathsf{Q}_1 + \mathsf{Q}_2 \wedge \mathsf{Q}_2). \tag{3.11}$$

In the rest of the paper we will construct bosonic (i.e. $N_1 = 0$) extended solutions with $N_0 = 0$ (i.e. the standard rank-2 case constructed above), with $N_0 = 1$ (i.e. rank-4 bosonic $R$-matrices) and with $N_0 = 2$ (i.e. rank-6 bosonic $R$-matrices), and we will find that $N_0 > 2$ is not possible. We will show that some $R$-matrices with $N_0 = 0$ and all of those that we construct with $N_0 = 1, 2$ admit also unimodular extensions (with $N_1 = N_0 + 1$). In the generic case, we prefer to rewrite (3.3) in terms of $\mathsf{e}_{\pm\mathsf{a}}$ for the even pairs in the extension with labels $\mathsf{a}, \mathsf{b} = 1, \ldots, N_0$, and $\mathsf{Q}_\mathsf{i}^\mathsf{I}$ with $\mathsf{i} = 1, 2$ for the odd pairs in the extension with labels $\mathsf{I}, \mathsf{J} = 1, \ldots, N_1$. The relations then read

$$\begin{aligned}
&[\mathsf{e}_{\pm\mathsf{a}}, \mathsf{e}] = 0, & &[\mathsf{h}, \mathsf{e}_{\pm\mathsf{a}}] = (\tfrac{1}{2} \pm \hat{\xi}_\mathsf{a})\mathsf{e}_{\pm\mathsf{a}}, & &[\mathsf{e}_{+\mathsf{a}}, \mathsf{e}_{-\mathsf{b}}] = \delta_{\mathsf{ab}}\mathsf{e}, & &[\mathsf{e}_{\pm\mathsf{a}}, \mathsf{e}_{\pm\mathsf{b}}] = 0, \\
&[\mathsf{Q}_\mathsf{i}^\mathsf{I}, \mathsf{e}] = 0, & &[\mathsf{h}, \mathsf{Q}_\mathsf{i}^\mathsf{I}] = \tfrac{1}{2}\mathsf{Q}_\mathsf{i}^\mathsf{I} - \varepsilon_{\mathsf{ij}}\check{\xi}^\mathsf{I}\,\mathsf{Q}_\mathsf{j}^\mathsf{I}, & &\{\mathsf{Q}_\mathsf{i}^\mathsf{I}, \mathsf{Q}_\mathsf{j}^\mathsf{J}\} = -i\delta^{\mathsf{IJ}}\delta_{\mathsf{ij}}\mathsf{e}, & &[\mathsf{Q}_\mathsf{i}^\mathsf{I}, \mathsf{e}_{\pm\mathsf{a}}] = 0,
\end{aligned} \tag{3.12}$$

and the $r$-matrix will be

$$r = \mathsf{h} \wedge \mathsf{e} + \sum_{\mathsf{a}=1}^{N_0} \mathsf{e}_{+\mathsf{a}} \wedge \mathsf{e}_{-\mathsf{a}} - \frac{i}{2} \sum_{\mathsf{I}=1}^{N_1} (\mathsf{Q}_1^\mathsf{I} \wedge \mathsf{Q}_1^\mathsf{I} + \mathsf{Q}_2^\mathsf{I} \wedge \mathsf{Q}_2^\mathsf{I}). \tag{3.13}$$

In order to respect the reality conditions of $\mathfrak{g}$ and, in our case, to generate real deformations of the $AdS_5 \times S^5$ background, we need to restrict the free parameters of the above relations to be real, $\hat{\xi}_\mathsf{a}, \check{\xi}^\mathsf{I} \in \mathbb{R}$. We refer to appendix A for more details.

# 4  Classification of the non-extended solutions in $\mathfrak{so}(2,4)$

Our first task is to obtain the classification of all rank-2 (i.e. non-extended with $N_0 = N_1 = 0$) Jordanian solutions of the conformal algebra[7] $\mathfrak{so}(2,4) \subset \mathfrak{psu}(2,2|4)$. In other words, we are after all the *inequivalent* choices of $\mathsf{h}$ and $\mathsf{e}$ among the elements of $\mathfrak{so}(2,4)$ that satisfy $[\mathsf{h},\mathsf{e}] = \mathsf{e}$. Two choices $\{\mathsf{h},\mathsf{e}\}$ and $\{\mathsf{h}',\mathsf{e}'\}$ are said to be equivalent if there exists an inner automorphism of $\mathfrak{so}(2,4)$ that relates them, i.e. if there exists an element $f \in SO(2,4)$ such that $\mathsf{h}' = f^{-1}\mathsf{h}f$ and $\mathsf{e}' = f^{-1}\mathsf{e}f$.

From the point of view of the classification of the deformations of $AdS_5 \times S^5$, modding out by inner automorphisms is justified by how the $R$-matrix enters the action of the deformed $\sigma$-model, see (A.6). In fact, $R$ appears in the linear operator $\mathcal{O} = 1 - \eta R_g \hat{d} : \mathfrak{g} \to \mathfrak{g}$, where $R_g = \mathrm{Ad}_g^{-1} R \, \mathrm{Ad}_g$. In the undeformed model, multiplication of the supercoset representative $g \in G$ from the left by a constant element $f \in G$ corresponds to an isometry of the target-space background. In the presence of the deformation, under a left multiplication we have invariance of the $\sigma$-model action (A.6) up to a possible change of the $R$-matrix itself as $R \to \mathrm{Ad}_f^{-1} R \, \mathrm{Ad}_f$. Therefore, the group of isometries is reduced to the subgroup of $G$ that leaves $R$ invariant (i.e. $R = \mathrm{Ad}_f^{-1} R \, \mathrm{Ad}_f$). Nevertheless, when $f$ does not correspond to an isometry because it does not leave the $R$-matrix (and therefore the action) invariant, multiplication by $f$ amounts to just a field redefinition of $g$, which is simply a different language to describe the same physics. Conversely, when two different deformations are generated by $R$ and $R'$ related as $R' = \mathrm{Ad}_f^{-1} R \, \mathrm{Ad}_f$, then it is enough to identify $g' = fg$ to conclude that the two deformations are physically equivalent. Notice that when $R' = \mathrm{Ad}_f^{-1} R \, \mathrm{Ad}_f$, then the relation between the Lie-algebra elements is implemented precisely by the adjoint $G$-action $T_I' = \mathrm{Ad}_f^{-1} T_I$, a fact which justifies the definition of equivalence for the choices $\{\mathsf{h},\mathsf{e}\}$ and $\{\mathsf{h}',\mathsf{e}'\}$.

According to the previous discussion, we must obtain all the embeddings of $\{\mathsf{h},\mathsf{e}\}$ in $\mathfrak{so}(2,4)$ modulo inner $SO(2,4)$ automorphisms. To do so, we will follow a strategy that was already used in [15] to classify all the inequivalent rank-4 unimodular bosonic $R$-matrices of $\mathfrak{so}(2,4)$. First, we notice that the 2-dimensional algebra generated by $\mathsf{h}$ and $\mathsf{e}$ is solvable. In [39] it was proved that all solvable subalgebras of $\mathfrak{so}(2,4)$ must be subalgebras of one of the maximal solvable subalgebras[8] of $\mathfrak{so}(2,4)$. The algebra $\mathfrak{so}(2,4)$ has two non-abelian maximal solvable subalgebras[9] which, following [15], we take as

$$\begin{aligned}
\mathfrak{s}_1 &= \mathrm{span}(p_i, J_{01} - J_{13}, J_{02} - J_{23}, J_{03}, J_{12}, D), \\
\mathfrak{s}_2 &= \mathrm{span}(p_0 + p_3, p_1, p_2, J_{01} - J_{13}, J_{02} - J_{23}, J_{12}, J_{03} - D, k_0 + k_3 + 2p_3) \,.
\end{aligned} \tag{4.1}$$

Let us stress that also the identification of $\mathfrak{s}_1$ and $\mathfrak{s}_2$ is provided only up to inner automorphisms of $\mathfrak{so}(2,4)$. That means that other choices are possible but they are all physically equivalent.

At this point we only need to find all the possible embeddings of the algebra generated by $\mathsf{h}$ and $\mathsf{e}$ in either $\mathfrak{s}_1$ or $\mathfrak{s}_2$, up to automorphisms generated by $SO(2,4)$. This task can be performed systematically by first identifying all possible embeddings of $\mathsf{e}$ up to inner automorphisms. The reason to single out this element first is that it is the only one that appears on the right-hand-side of the commutation relation. All such possible embeddings of $\mathsf{e}$ were already worked out in [15] and we recap them in Table 5.

---

[7] The algebra spanned by $\mathsf{h},\mathsf{e}$ is non-compact and thus nor $\mathsf{h}$ nor $\mathsf{e}$ can be an element of $\mathfrak{so}(6) \subset \mathfrak{psu}(2,2|4)$. Nevertheless, as mentioned in the previous section, $\mathsf{h}$ and $\mathsf{e}$ can be shifted by $\mathfrak{so}(6)$ elements and the classical Yang-Baxter equation will still hold.

[8] See the corollary at the end of section II.B of [39].

[9] See the results of section III.D of [39] for the group $SU(2,2)$ which is locally isomorphic to $SO(2,4)$, as well as [40]. Notice that we are ignoring the maximal solvable subalgebra $\mathfrak{s}_0$ because it is abelian. Moreover, following [15], we swap the definition of $\mathfrak{s}_1$ and $\mathfrak{s}_2$ compared to [39].

|  | $\mathfrak{s}_1$ |  | $\mathfrak{s}_2$ |
|---|---|---|---|
| (1) | $p_1$ | (1) | $p_1$ |
| (2) | $J_{02} - J_{23}$ | (2) | $p_0 + p_3$ |
| (3) | $p_1 + J_{02} - J_{23}$ | (3) | $a\,p_1 + b\,p_2 + J_{01} - J_{13}$ |
| (4) | $p_0$ | | |
| (5) | $p_3$ | | |
| (6) | $p_0 + p_3$ | | |
| (7) | $p_0 - p_3 + J_{01} - J_{13}$ | | |

Table 5: All the possible inequivalent embeddings of $\mathsf{e}$ in the two non-abelian maximal solvable subalgebras of $\mathfrak{so}(2,4)$, up to inner $SO(2,4)$ automorphisms. In option (3) of $\mathfrak{s}_2$, $a$ and $b$ are two real parameters.

After fixing a choice for $\mathsf{e}$, it is a matter of imposing the commutation relation $[\mathsf{h}, \mathsf{e}] = \mathsf{e}$ to find $\mathsf{h}$ as well. After doing so, one must act again with inner $SO(2,4)$ automorphisms in order to remove as many free parameters as possible, and therefore identify all the *inequivalent* embeddings of $\mathsf{h}$. In the following, when saying that we use an automorphism generated by $x \in \mathfrak{so}(2,4)$ we mean that we implement the transformation $\mathsf{h} \to e^{-x}\mathsf{h}\,e^x$ and $\mathsf{e} \to e^{-x}\mathsf{e}\,e^x$. Given that the procedure assumes that the choice for $\mathsf{e}$ is fixed, we will only consider transformations that leave $\mathsf{e}$ invariant ($e^{-x}\mathsf{e}\,e^x = \mathsf{e}$) or at most that they rescale it by an overall factor ($e^{-x}\mathsf{e}\,e^x = c\,\mathsf{e}$) because that can be reabsorbed by the redefinition of the deformation parameter in the action.[10] In general, note that in $\mathsf{h}$ we can never remove contributions from the last three generators of $\mathfrak{s}_1$ and $\mathfrak{s}_2$, because these generators never appear on the right-hand side of the commutation relations of these subalgebras of $\mathfrak{so}(2,4)$.

We will describe in some detail the calculations for the first example, and we will summarise as briefly as possible the remaining ones.

## 4.1  Embeddings in $\mathfrak{s}_1$

### (1)  $\mathsf{e} = p_1$

In order to identify $\mathsf{h}$ one starts from the parameterisation of a generic element of $\mathfrak{s}_1$ and one takes for example $\mathsf{h} = \alpha^i p_i + \beta(J_{01} - J_{13}) + \gamma(J_{02} - J_{23}) + \delta J_{03} + \epsilon J_{12} + \lambda D$ with all the coefficients beaing generic real parameters. After imposing $[\mathsf{h}, \mathsf{e}] = \mathsf{e}$, one finds that some of the parameters need to be fixed to special values. In particular we find $\mathsf{h} = D + \alpha^i p_i + \gamma(J_{02} - J_{23}) + \delta J_{03}$. At this point we can act with inner $SO(2,4)$ automorphisms to further reduce the number of physical parameters. This is what we will explain in the following.

First of all, if $\delta \neq \pm 1$, then all contributions with $p_i$ in $\mathsf{h}$ can be removed by acting with an automorphism generated by $x = c^i p_i$ with $c^i \in \mathbb{R}$. In fact, after noticing that $e^{-x}\mathsf{e}\,e^x = \mathsf{e}$, one finds that

$$e^{-x}\mathsf{h}\,e^x = D + (c^0 - \gamma c^2 - \delta c^3 + \alpha^0)p_0 + (c^1 + \alpha^1)p_1 + (c^2 + \alpha^2 - \gamma c^0 + \gamma c^3)p_2$$
$$+ (c^3 + \alpha^3 - \gamma c^2 - \delta c^0)p_3 + \gamma(J_{02} - J_{23}) + \delta J_{03}, \tag{4.2}$$

so that to remove the contribution of $p_1$ it is enough to set $c_1 = -\alpha_1$. Removing the contributions with $p_0, p_2, p_3$ is more delicate: one needs to impose a system of 3 linear equations for the 3 unknowns $c^0, c^2, c^3$. The determinant of the matrix associated to this linear system is $\det = 1 - \delta^2$.

---

[10]This transformation, in fact, leaves $[\mathsf{h}, \mathsf{e}] = \mathsf{e}$ invariant. One could consider also $\mathsf{e} \to \mathsf{e} + c\,\mathsf{h}$, but it would generate new (non-physical) parameters rather than reabsorbing them.

Therefore, if we assume that $\delta \neq \pm 1$ we can remove all contributions with $p_i$ as anticipated above. At this point, an automorphism generated by $x = c(J_{02} - J_{23})$ is able to remove also $J_{02} - J_{23}$, at the extra condition that $\delta \neq 0$. Notice that if $\delta = 0$ and $\gamma \neq 0$, then we can still set it to $\pm 1$ by acting with $x = cJ_{03}$.

If $\delta = 1$, instead, after removing $p_1$ by $x = c^1 p_1$, we can also remove $J_{02} - J_{23}$ by $x = c(J_{02} - J_{23})$ because now we are sure to satisfy the condition $\delta \neq 0$. A further action with $x = c^i p_i$ (with $c^1 = 0$) allows us to remove $p_2$ and $p_0 - p_3$ from $\mathsf{h}$, but not necessarily $p_0 + p_3$. If not 0, the coefficient in front of $p_0 + p_3$ can be set to $\pm 1$ by $x = cD$. If $\delta = -1$ then the reasoning is similar, but with the role of $p_0 - p_3$ and $p_0 + p_3$ interchanged.

To recap, we have the following 3 inequivalent possibilities:

$$
\begin{aligned}
&1.1.a: \quad \mathsf{h} = D + \delta J_{03}, \\
&1.1.b: \quad \mathsf{h} = D + \gamma(J_{02} - J_{23}), \quad \text{with } \gamma^2 = 1, \\
&1.1.c: \quad \mathsf{h} = D + \alpha(p_0 + \delta p_3) + \delta J_{03}, \quad \text{with } \alpha^2 = \delta^2 = 1.
\end{aligned}
\tag{4.3}
$$

## (2)  $\mathbf{e = J_{02} - J_{23}}$

Generically we may have $\mathsf{h} = \alpha^+(p_0 + p_3) + \alpha^1 p_1 + \beta(J_{01} - J_{13}) + \gamma(J_{02} - J_{23}) + \lambda D - J_{03}$. First, we can remove both $J_{01} - J_{13}$ and $J_{02} - J_{23}$ by $x = c^1(J_{01} - J_{13}) + c^2(J_{02} - J_{23})$. If $\lambda \neq 0$ and $\lambda \neq -1$, then we can also remove $p_0 + p_3$ and $p_1$ by $x = c^+(p_0 + p_3) + c^1 p_1$. If $\lambda = 0$ we can remove $p_0 + p_3$ but not necessarily $p_1$, and the coefficient of the latter (if not 0) can be set to $\pm 1$ by $x = cD$. Similarly, if $\lambda = -1$ then we can remove $p_1$ but not necessarily $p_0 + p_3$, and the coefficient of the latter can also be set to $\pm 1$.

To recap, we have the following 3 inequivalent possibilities:

$$
\begin{aligned}
&1.2.a: \quad \mathsf{h} = -J_{03} + \lambda D, \\
&1.2.b: \quad \mathsf{h} = -J_{03} + \alpha p_1, \quad \text{with } \alpha^2 = 1, \\
&1.2.c: \quad \mathsf{h} = -J_{03} - D + \alpha(p_0 + p_3), \quad \text{with } \alpha^2 = 1.
\end{aligned}
\tag{4.4}
$$

## (3)  $\mathbf{e = p_1 + J_{02} - J_{23}}$

Generically we may have $\mathsf{h} = D - J_{03} + \alpha^+(p_0 + p_3) + \alpha^1 p_1 + \beta(p_2 + J_{01} - J_{13}) + \gamma(J_{02} - J_{23})$. First, we can act with an automorphism generated by $x = c(p_2 + J_{01} - J_{13})$ to remove the contribution proportional to $\beta$. Similarly, acting with $x = cp_1$ will remove $p_1$, with $x = c(J_{02} - J_{23})$ will remove $J_{02} - J_{23}$ and with $x = c(p_0 + p_3)$ will remove $p_0 + p_3$.

In this example, therefore, there is only one possibility, namely

$$
\mathsf{h} = D - J_{03}.
\tag{4.5}
$$

## (4)  $\mathbf{e = p_0}$

We must start from $\mathsf{h} = D + \epsilon J_{12} + \alpha^i p_i$. Acting with $x = c^i p_i$ we can always remove all the $p_i$. The contribution with $J_{12}$, instead, cannot be removed and we have only one possibility

$$
\mathsf{h} = D + \epsilon J_{12}.
\tag{4.6}
$$

**(5)  e = p₃**

As in the previous example, we have $\mathsf{h} = D + \epsilon J_{12} + \alpha^i p_i$ and all contributions with $p_i$ can be removed with $x = c^i p_i$. We therefore have only one possibility

$$\mathsf{h} = D + \epsilon J_{12}. \qquad (4.7)$$

**(6)  e = p₀ + p₃**

In general we have $\mathsf{h} = \alpha^i p_i + \beta(J_{01} - J_{13}) + \gamma(J_{02} - J_{23}) + \delta J_{03} + \epsilon J_{12} + (\delta + 1)D$. If at least one of the two parameters $\delta, \epsilon$ is non-vanishing (i.e. if $\delta^2 + \epsilon^2 \neq 0$) then we can remove the contributions with $J_{01} - J_{13}, J_{02} - J_{23}$ with $x = c^1(J_{01} - J_{13}) + c^2(J_{02} - J_{23})$. If we further assume that $\delta \neq -\frac{1}{2}$ and that $(\delta + 1)^2 + \epsilon^2 \neq 0$ then we can also remove all $p_i$ with $x = c^i p_i$.

If instead $\delta = -\frac{1}{2}$ we can remove all $p_i$ except the combination $p_0 - p_3$, whose coefficient may be set to $\pm 1$ with $x = cD$.

Another scenario is $\delta = -1$ and $\epsilon = 0$. We can remove $p_0, p_3$ but not $p_1, p_2$ with $x = c^i p_i$. Nevertheless, if both $p_1, p_2$ are present, we can remove one of them by $x = cJ_{12}$, and set the coefficient of the remaining one to $\pm 1$ by $x = cD$.

Finally, if $\delta = 0$ and $\epsilon = 0$ then we can act with $x = cJ_{12}$ to remove, for example, $J_{02} - J_{23}$. After doing that, we can always remove all $p_i$ by $x = c^i p_i$. At this point, it is possible to rescale the coefficient of $J_{01} - J_{13}$ by $x = cJ_{03}$.

To summarise, we have in total 4 possibilities:

$$
\begin{aligned}
&1.6.a: \quad \mathsf{h} = (\delta + 1)D + \delta J_{03} + \epsilon J_{12}, \\
&1.6.b: \quad \mathsf{h} = \tfrac{1}{2}(D - J_{03}) + \epsilon J_{12} + \alpha(p_0 - p_3), \ \text{with } \alpha^2 = 1, \\
&1.6.c: \quad \mathsf{h} = -J_{03} + \alpha p_1, \ \text{with } \alpha^2 = 1, \\
&1.6.d: \quad \mathsf{h} = D + \alpha(J_{01} - J_{13}), \ \text{with } \alpha^2 = 1.
\end{aligned}
\qquad (4.8)
$$

**(7)  e = p₀ − p₃ + J₀₁ − J₁₃**

We start from $\mathsf{h} = 2D - J_{03} + \alpha^+(p_0 + p_3) + \alpha^2 p_2 + \beta(J_{01} - J_{13}) + \gamma(J_{02} - J_{23})$. We can remove $J_{01} - J_{13}, J_{02} - J_{23}$ with $x = c^1(J_{01} - J_{13}) + c^2(J_{02} - J_{23})$, $p_2$ with $x = cp_2$ and $p_0 + p_3$ with $x = c(p_0 + p_3)$. Therefore, we only have one possibility

$$\mathsf{h} = 2D - J_{03}. \qquad (4.9)$$

## 4.2   Embeddings in 𝔰₂

**(1)  e = p₁**

We start from the parameterisation of a generic element of $\mathfrak{s}_2$, namely $\mathsf{h} = \alpha^+(p_0 + p_3) + \alpha^1 p_1 + \alpha^2 p_2 + \beta(J_{01} - J_{13}) + \gamma(J_{02} - J_{23}) + \delta(J_{03} - D) + \epsilon J_{12} + \lambda(k_0 + k_3 + 2p_3)$. After imposing the commutation relation $[\mathsf{h}, \mathsf{e}] = \mathsf{e}$ we find $\mathsf{h} = D - J_{03} + \alpha^+(p_0 + p_3) + \alpha^1 p_1 + \alpha^2 p_2 + \gamma(J_{02} - J_{23})$. We can remove the contribution with $J_{02} - J_{23}$ by acting with an automorphism generated by $x = c(J_{02} - J_{23})$. After that, we can remove all $p_i$ by $x = c^+(p_0 + p_3) + c^1 p_1 + c^2 p_2$. Therefore, we simply have

$$\mathsf{h} = D - J_{03}. \qquad (4.10)$$

Actually this solution is the same found in case (1) of the $\mathfrak{s}_1$ embedding. This simply means that the algebra that we are considering is a subalgebra of both $\mathfrak{s}_1$ and $\mathfrak{s}_2$.

## (2) $\;$ $\mathsf{e} = p_0 + p_3$

We have $\mathsf{h} = \frac{1}{2}(D - J_{03}) + \alpha^+(p_0 + p_3) + \alpha^1 p_1 + \alpha^2 p_2 + \beta(J_{01} - J_{13}) + \gamma(J_{02} - J_{23}) + \epsilon J_{12} + \lambda(k_0 + k_3 + 2p_3)$. We can remove the contributions with $J_{01} - J_{13}, J_{02} - J_{23}$ by $x = c^1(J_{01} - J_{13}) + c^2(J_{02} - J_{23})$, and similarly the contributions of $p_0 + p_3, p_1, p_2$. Therefore we have

$$\mathsf{h} = \tfrac{1}{2}(D - J_{03}) + \epsilon J_{12} + \lambda(k_0 + k_3 + 2p_3). \tag{4.11}$$

## (3) $\;$ $\mathsf{e} = ap_1 + bp_2 + J_{01} - J_{13}$

Here we need to distinguish different cases. If $a^2 - b^2 + 1 \neq 0$ or if $ab \neq 0$ then we have $\mathsf{h} = D - J_{03} + \alpha^+(p_0 + p_3) + (a\beta + b\gamma)p_1 + \alpha^2 p_2 + \beta(J_{01} - J_{13}) + \gamma(J_{02} - J_{23})$. Now acting with $x = c^+(p_0 + p_3) + c^1 p_1 + c^2 p_2 + c_\beta(J_{01} - J_{13}) + c_\gamma(J_{02} - J_{23})$ we can remove all free parameters and have just

$$2.3.a : \;\; \mathsf{h} = D - J_{03}. \tag{4.12}$$

On the other hand, if $a = 0$ and $b = \pm 1$ (which is the only real solution of the system $a^2 - b^2 + 1 = 0$ and $ab = 0$) then we have $\mathsf{h} = D - J_{03} + \alpha^+(p_0 + p_3) \pm \gamma p_1 + \alpha^2 p_2 + \beta(J_{01} - J_{13}) + \gamma(J_{02} - J_{23}) \mp 2\lambda J_{12} + \lambda(k_0 + k_3 + 2p_3)$. At this point we can act with $x = c^+(p_0 + p_3) + c^1 p_1 + c^2 p_2 + c_\beta(J_{01} - J_{13}) + c_\gamma(J_{02} - J_{23})$ to remove several parameters and be left with only

$$2.3.b : \;\; \mathsf{h} = D - J_{03} \mp 2\lambda J_{12} + \lambda(k_0 + k_3 + 2p_3). \tag{4.13}$$

# 5 $\;$ Classification of the (bosonic) extended solutions in $\mathfrak{so}(2,4)$

The reasoning followed to classify the rank-2 solutions in $\mathfrak{so}(2,4)$ can be applied also to find extended (i.e. higher rank) bosonic Jordanian $R$-matrices. In fact, according to the commutation relations given in (3.3) or (3.12), the $N$-extended algebra is also solvable, and if we want it to be a subalgebra of $\mathfrak{so}(2,4)$ it must again be a subalgebra of either $\mathfrak{s}_1$ or $\mathfrak{s}_2$. Here we use this observation to classify the extended (higher-rank) Jordanian $R$-matrices. The strategy is to start from the classification of the rank-2 solutions of the previous section and construct elements $\mathsf{e}_{\pm a}$ that satisfy (3.12) with a given $\mathsf{h}$ and $\mathsf{e}$ up to inner $\mathfrak{so}(2,4)$ automorphisms that now leave *both* $\mathsf{h}$ and $\mathsf{e}$ invariant.

It turns out that for most choices of $\mathsf{e} \in \mathfrak{s}_i$ it is not possible to identify two elements $\mathsf{e}_+, \mathsf{e}_- \in \mathfrak{s}_i$ that commute with $\mathsf{e}$ and that satisfy $[\mathsf{e}_+, \mathsf{e}_-] = \mathsf{e}$. We therefore conclude that in those cases it is not possible to construct bosonic extended solutions. We find that it is possible to construct such solutions only when $\mathsf{e} = p_0 + p_3$. This option shows up both in $\mathfrak{s}_1$ and $\mathfrak{s}_2$, and because it is a subset of the rank-2 solutions admitting a unimodular extension of Table 1, we will refer to the names used in that table. Table 1 also summarises the inner $\mathfrak{so}(2,4)$ automorphisms that we can exploit. When constructing the extended solutions we will refer to the names used in the summary of results of section 2.

## 5.1 Bosonic extensions in $\mathfrak{s}_1$

### $N_0 = 1$

Let us first try to construct an extension with $N_0 = 1$. We start from

$$\mathsf{e}_\pm = \alpha^i_\pm p_i + \beta_\pm (J_{01} - J_{13}) + \gamma_\pm (J_{02} - J_{23}) + \delta_\pm (D + J_{03}) + \epsilon_\pm J_{12}, \tag{5.1}$$

since these commute with $\mathsf{e} = p_0 + p_3$. After imposing by brute force the relations in (3.12) with $\mathsf{h}$ given in $R_1$ of Table 1 (i.e. case 1.6.$a$ of (4.8)), we find that for $\hat{\xi}$ generic we can have the following solution[11]

$$\delta = \hat{\xi} - \tfrac{1}{2}, \quad \epsilon = 0, \qquad (\implies \mathsf{h} = (1+\delta)D + \delta J_{03}),$$
$$\mathsf{e}_+ = \bar{\mathsf{e}}_+ \equiv a\, p_1 + b\, p_2, \qquad \mathsf{e}_- = \bar{\mathsf{e}}_- \equiv c\,(J_{01} - J_{13}) + d\,(J_{02} - J_{23}), \tag{5.2}$$

where the free (real) parameters are constrained to satisfy

$$a\,c + b\,d = 1. \tag{5.3}$$

We may actually act with an automorphism generated by a $J_{12}$ rotation and remove the dependence on one of the above parameters. Without loss of generality[12] we may set for example $c = 0$, so that the quadratic condition reduces to $d = b^{-1}$. At this point, we can act with an automorphism generated by $D + J_{03}$ which allows us to set $b = \pm 1$. Taking into account that the transformation $\mathsf{e}_\pm \to c^{\pm 1}\mathsf{e}_\pm$ for any $c \in \mathbb{R}$ is an automorphism of the algebra and leaves the $R$-matrix invariant, we can effectively set $b = 1$. This is the solution $R_7$ in Table 4.

When $\hat{\xi}$ takes some special values, the solutions for $\mathsf{e}_\pm$ can be slightly more generic. First we find[13]

$$\hat{\xi} = \tfrac{1}{6}, \quad \delta = -\frac{1}{3}, \quad \epsilon = 0, \quad (\implies \mathsf{h} = \tfrac{2}{3}D - \tfrac{1}{3}J_{03}),$$
$$\mathsf{e}_- = f(p_0 - p_3) + \bar{\mathsf{e}}_-, \qquad \mathsf{e}_+ = \bar{\mathsf{e}}_+, \tag{5.4}$$

still subject to the constraint (5.3) and with the parameter $f \in \mathbb{R}$ free. As done previously, we can set $c = 0$ by $J_{12}$, then $b = \pm 1$ by $D + J_{03}$, and then send $\mathsf{e}_\pm \to b^{\pm 1}\mathsf{e}_\pm$ yielding $R_8$ in Table 4.

Second, we also have the option

$$\hat{\xi} = 0, \qquad \delta = -\frac{1}{2}, \qquad \epsilon = 0, \quad (\implies \mathsf{h} = \tfrac{1}{2}(D - J_{03})),$$
$$\mathsf{e}_+ = a_+ p_1 + b_+ p_2 + c_+(J_{01} - J_{13}) + d_+(J_{02} - J_{23}),$$
$$\mathsf{e}_- = a_- p_1 + b_- p_2 + c_-(J_{01} - J_{13}) + d_-(J_{02} - J_{23}), \tag{5.5}$$

with the more general constraint

$$a_+ c_- + b_+ d_- - a_- c_+ - b_- d_+ = 1. \tag{5.6}$$

---

[11]Notice that (3.12) is symmetric under the transformation $\hat{\xi}_\mathsf{a} \to -\hat{\xi}_\mathsf{a}$ and $\mathsf{e}_{\pm\mathsf{a}} \to \pm\mathsf{e}_{\mp\mathsf{a}}$, so that in principle we also have the solution with $\delta = -\hat{\xi} - \tfrac{1}{2}$, $\epsilon = 0$ and $\mathsf{e}_+ = \bar{\mathsf{e}}_-$, $\mathsf{e}_- = -\bar{\mathsf{e}}_+$. We do not consider these as independent solutions because they only amount to a redefinition of the basis of the algebra. In fact, the $R$-matrix is unchanged under the transformation $\mathsf{e}_{\pm\mathsf{a}} \to \pm\mathsf{e}_{\mp\mathsf{a}}$.

[12]Because of the condition (5.3) we cannot have at the same time $c = 0$ and $d = 0$.

[13]As already remarked, the relations in (3.12) are symmetric under $\hat{\xi}_\mathsf{a} \to -\hat{\xi}_\mathsf{a}$ and $\mathsf{e}_{\pm\mathsf{a}} \to \pm\mathsf{e}_{\mp\mathsf{a}}$. Therefore we have also solutions obtained by this transformation, namely with $\hat{\xi} = -\tfrac{1}{2}$ and $\hat{\xi} = -\tfrac{1}{6}$. These are not independent solutions, see footnote 11. In principle, we also find additional solutions with either $\mathsf{e}_+$ or $\mathsf{e}_-$ containing a contribution proportional to $p_0 + p_3$, but this can be shifted away by an automorphism, so that the solution reduces to a special case of $R_7$.

For this constraint to admit a solution, notice that either $e_+$ or $e_-$ must have a $p_i$ with non-vanishing coefficient, and the other must have $J_{0i} - J_{i3}$ with non vanishing coefficient. For definiteness, let us assume that $a_+ \neq 0$ and $c_- \neq 0$. The other options are obtained by applying the symmetries $e_\pm \to \pm e_\pm$ or $1 \leftrightarrow 2$ of the spatial indices 1,2, that give rise to equivalent solutions. As we can see in Table 1, in this case $h$ and $e$ are invariant not only under the action of $D + J_{03}$ and $J_{12}$ but also $p_0 - p_3$ and $k_0 + k_3$. It should be possible to use the last 3 of these generators to set $c_+ = b_+ = b_- = 0$, so that the quadratic constraint reduces to $c_- = a_+^{-1}$. After that, using $D + J_{03}$ one could set $a_+ = \pm 1$. In other words, we believe it is possible to reduce this case to

$$\hat{\xi} = 0, \qquad \delta = -\frac{1}{2}, \qquad \epsilon = 0, \quad (\implies \quad h = \tfrac{1}{2}(D - J_{03})),$$
$$e_+ = \alpha p_1 + d_+ (J_{02} - J_{23}), \tag{5.7}$$
$$e_- = a_- p_1 + \alpha (J_{01} - J_{13}) + d_- (J_{02} - J_{23}), \qquad \text{with } \alpha^2 = 1.$$

However, it is quite subtle to make sure that setting $c_+ = b_+ = b_- = 0$ is possible for all values of the initial parameters $a_\pm, b_\pm, c_\pm, d_\pm$. In fact, one may worry about possible singularities for special values of these parameters. The actions of the generators $J_{12}, p_0 - p_3, k_0 + k_3$ mix non-trivially, and this makes the analysis more complicated. For this reason, the solution $R_9$ in Table 4 is presented without the maximal simplification by automorphisms. We simply set $c_+ = 0$ by means of $k_0 + k_3$, then set $a_+ = \pm 1$ thanks to $D + J_{03}$ and finally send $e_\pm \to a_+^{\pm 1} e_\pm$.

The calculations to identify the possible $N_0 = 1$ extensions of $R_2$ (i.e. the case 1.6.b in (4.8)) are similar to the above ones (when setting $\hat{\xi} = 0, \delta = -\frac{1}{2}$). However, no solution is possible because $p_0 - p_3$ does not commute with $J_{01} - J_{13}$ and $J_{02} - J_{23}$, and therefore $R_2$ does not admit a bosonic extension. Also the calculation for $R_3$ (i.e. case 1.6.c) is similar to the ones above (now setting $\delta = -1$) and we can obtain the solution $R_{10}$ of Table 4 by borrowing the extension $R_7$ (before removing any parameter by conjugation) at the condition that we further set $c = 0$ in $\bar{e}_-$ (because of the extra contribution of $p_1$ in $h$). Similarly, $R_4$ (i.e. case 1.6.d) admits an extension as in $R_7$ at the condition of further setting $a = 0$ in $\bar{e}_+$, giving rise to the solution $R_{11}$. Note that, both in the case of $R_{10}$ and $R_{11}$, $J_{12}$ and $D + J_{03}$ do not commute with $h$. Therefore, one can not use these to remove and rescale parameters.

## $N_0 = 2$

Using the above results, we can check that it is possible to construct also extended bosonic Jordanian solutions with $N_0 = 2$. In particular, for generic $\hat{\xi}$ we can start from $R_7$ and construct the solution[14]

$$R_{15} : \delta = \hat{\xi} - \tfrac{1}{2}, \quad \epsilon = 0, \qquad (\implies \quad h = (\hat{\xi} + \tfrac{1}{2})D + (\hat{\xi} - \tfrac{1}{2})J_{03}),$$
$$e_{+a} = \bar{e}_{+a} \equiv a_a p_1 + b_a p_2, \qquad e_{-a} = \bar{e}_{-a} \equiv c_a(J_{01} - J_{13}) + d_a(J_{02} - J_{23}), \tag{5.8}$$

with the simultaneous conditions

$$a_2 c_1 + b_2 d_1 = 0, \qquad a_1 c_2 + b_1 d_2 = 0, \qquad a_a c_a + b_a d_a = 1, \quad a = 1, 2. \tag{5.9}$$

Using the automorphism generated by a $J_{12}$ rotation we can set for example $c_1 = 0$ which then implies $b_2 = 0$ and $d_1 = 1/b_1, c_2 = 1/a_2, d_2 = -a_1/(b_1 a_2)$. Moreover, using $D + J_{03}$ we can set $b_1 = \pm 1$ and finally taking into account that also the transformation $e_{\pm a} \to c^{\pm 1} e_{\pm a}$ is an

---

[14]In principle each pair $\{e_{+a}, e_{-a}\}$ may come with its own coefficient $\hat{\xi}_a$, but $\delta$ is already constrained to be $\delta = \pm \hat{\xi} - \frac{1}{2}$ which implies that $\hat{\xi}_1 = \hat{\xi}_2 = \hat{\xi}$. We could in fact combine different solutions related by $\hat{\xi} \to -\hat{\xi}$ but they would be equivalent to those that we write here.

automorphism of the algebra (and leaves the $R$-matrix invariant) we can simply set $a_2 = 1$. This gives the solution $R_{13}$ of section 2.

It is not possible, instead, to construct an extended solution from $R_8$ because $p_0 - p_3$ does not commute with $J_{01} - J_{13}$ nor $J_{02} - J_{23}$. Therefore we cannot construct two linearly independent elements $\mathsf{e}_{-1}, \mathsf{e}_{-2}$ unless they both have a vanishing coefficient in front of $p_0 - p_3$ ($f_\mathsf{a} = 0$), which then reduces to the previous solution. Finally, from $R_9$ we can construct the following extended solution

$$R_{14} : \hat{\xi} = 0, \qquad \delta = -\frac{1}{2}, \qquad \epsilon = 0, \qquad ( \implies \ \mathsf{h} = \tfrac{1}{2}(D - J_{03})),$$
$$\mathsf{e}_{\pm\mathsf{a}} = a_{\pm\mathsf{a}}p_1 + b_{\pm\mathsf{a}}p_2 + c_{\pm\mathsf{a}}(J_{01} - J_{13}) + d_{\pm\mathsf{a}}(J_{02} - J_{23}),$$
(5.10)

with the 6 constraints

$$a_{\pm 1}c_{\pm 2} + b_{\pm 1}d_{\pm 2} - b_{\pm 2}d_{\pm 1} - a_{\pm 2}c_{\pm 1} = 0,$$
$$a_{\pm 1}c_{\mp 2} + b_{\pm 1}d_{\mp 2} - b_{\mp 2}d_{\pm 1} - a_{\mp 2}c_{\pm 1} = 0,$$
$$a_{+\mathsf{a}}c_{-\mathsf{a}} + b_{+\mathsf{a}}d_{-\mathsf{a}} - a_{-\mathsf{a}}c_{+\mathsf{a}} - b_{-\mathsf{a}}d_{+\mathsf{a}} = 1, \quad \mathsf{a} = 1, 2.$$
(5.11)

These are all the options for extended solutions with $N_0 = 2$. It is not possible to construct them from $R_3$ and $R_4$ (or equivalently $R_{10}$ and $R_{11}$) because the extra condition of not having either $J_{01} - J_{13}$ or $p_1$ in $\mathsf{e}_{\pm\mathsf{a}}$ implies that there would not be enough linearly-independent vectors to construct the full solution.

For a similar reason, it is obviously not possible to construct extended solutions with $N_0 > 2$. There would not be enough linearly-independent vectors to construct the pairs $\{\mathsf{e}_{+\mathsf{a}}, \mathsf{e}_{-\mathsf{a}}\}$.

## 5.2  Bosonic extensions in $\mathfrak{s}_2$

### $N_0 = 1$

To understand whether we can construct a bosonic extension in this case, we proceed as before starting from

$$\mathsf{e}_\pm = \alpha_\pm^+(p_0+p_3)+\alpha_\pm^1 p_1+\alpha_\pm^2 p_2+\beta_\pm(J_{01}-J_{13})+\gamma_\pm(J_{02}-J_{23})+\epsilon_\pm J_{12}+\lambda_\pm(k_0+k_3+2p_3), \quad (5.12)$$

that commute with $\mathsf{e}$. We notice that this is a slight modification of the calculation done for the embedding in $\mathfrak{s}_1$: first there is no combination $p_0 - p_3$, and second there is an additional contribution of $k_0 + k_3 + 2p_3$ both in $\mathsf{e}_\pm$ and in $\mathsf{h}$. In order to find a bosonic extension to this solution that is new compared to what found in $\mathfrak{s}_1$, we must therefore have $\lambda \neq 0$ in $\mathsf{h}$ or $\lambda_+ \neq 0$ in $\mathsf{e}_+$ or $\lambda_-$ in $\mathsf{e}_-$ (or more than one of these possibilities simultaneously). We find that it is possible to construct extended solutions of this kind at $\hat{\xi} = 0$ if we set $\lambda = \pm\frac{\epsilon}{2}$ in (4.11) and if we take

$$\mathsf{e}_+ = a_+ p_1 + b_+ p_2 \mp b_+(J_{01} - J_{13}) \pm a_+(J_{02} - J_{23}),$$
$$\mathsf{e}_- = a_- p_1 + b_- p_2 \mp b_-(J_{01} - J_{13}) \pm a_-(J_{02} - J_{23}),$$
(5.13)

with the condition

$$a_+b_- - a_-b_+ = \mp\tfrac{1}{2},$$
(5.14)

where the signs are correlated to the choice of sign in $\lambda = \pm\frac{\epsilon}{2}$. Note that to have a genuinely new extension (compared to the embedding in $\mathfrak{s}_1$) we must assume $\epsilon \neq 0$. Although both $k_0 + k_3 - p_0 + p_3$ and $J_{12}$ leave $\mathsf{h}$ and $\mathsf{e}$ invariant, we can remove only one parameter by their actions (because it turns out that there is only one non-trivial parameter when applying the two actions simultaneously) and we decide to set $b_+ = 0$. Then the constraint is solved just by $b_- = \mp\frac{1}{2a_+}$. Nevertheless, we can always shift $\mathsf{e}_-$ by a quantity proportional to $\mathsf{e}_+$, given that at

the level of the $R$-matrix the contribution of this extra shift drops out because of antisymmetry of $R$.[15] Therefore, the contribution proportional to $a_-$ produces no effect, and we can just set $a_- = 0$. This is the solution $R_{12}$ in Table 4.

In this case it is not possible to construct bosonic extended solutions with $N_0 > 1$. There are simply not enough parameters to satisfy all needed conditions.

# 6 Classification of the unimodular solutions in $\mathfrak{psu}(2,2|4)$

In this section we address the question of whether it is possible to extend the bosonic Jordanian solutions constructed above to obtain unimodular solutions. As recalled in section 3, we should add $N_1$ pairs of odd generators from $\mathfrak{psu}(2,2|4)$ with $N_1 = N_0 + 1$, where $N_0$ is the number of extra bosonic pairs of generators in the bosonic extension.

We will construct odd elements as linear combinations of the supercharges of $\mathfrak{psu}(2,2|4)$

$$\mathsf{Q}_\mathsf{i}^\mathsf{l} = q_\mathsf{i}^\mathsf{l} \cdot Q + \overline{q}_\mathsf{i}^\mathsf{l} \cdot \overline{Q} + s_\mathsf{i}^\mathsf{l} \cdot S + \overline{s}_\mathsf{i}^\mathsf{l} \cdot \overline{S}. \tag{6.1}$$

Here we are using a simplifying notation where the dot product means that the indices of the supercharges are assumed to be in their canonical position $Q_{\alpha a}, \overline{Q}^{\dot{\alpha} a}, S_\alpha{}^a, \overline{S}^{\dot{\alpha}}{}_a$ and are contracted by complex coefficients $q, \overline{q}, s, \overline{s}$ (where we omit the obvious $\mathsf{l}, \mathsf{i}$ indices for simplicity) with Lorentz and $\mathcal{R}$-symmetry indices in appropriate positions.[16] Importantly, we will demand that the odd elements $\mathsf{Q}_\mathsf{i}^\mathsf{l}$ satisfy the standard reality conditions of $\mathfrak{psu}(2,2|4)$, namely $(\mathsf{Q}_\mathsf{i}^\mathsf{l})^\dagger + H\mathsf{Q}_\mathsf{i}^\mathsf{l} H^{-1} = 0$, see appendix B for more details. Given that the supercharges that we are using for the superalgebra basis are not real and satisfy (B.25) instead, this implies that the complex coefficients $q, \overline{q}, s, \overline{s}$ must be such that

$$(q^{\alpha a})^* = -\epsilon^{\alpha\beta}\delta^{ab}\overline{q}_{\beta b}, \qquad (s^\alpha{}_a)^* = -\epsilon^{\alpha\beta}\delta_{ab}\overline{s}_\beta{}^b, \tag{6.2}$$

where the star is the complex conjugation.[17] In total we therefore have 32 real coefficients for each $\mathsf{Q}_\mathsf{i}^\mathsf{l}$. Alternatively, we may define the real supercharges[18]

$$\mathcal{Q}_\pm^{\alpha a} \equiv \overline{Q}^{\alpha a} \pm \epsilon^{\alpha\beta}\delta^{ab}Q_{\beta b}, \qquad \mathcal{S}_{\pm a}^\alpha \equiv \overline{S}^\alpha{}_a \pm \epsilon^{\alpha\beta}\delta_{ab}S_\beta{}^b, \tag{6.3}$$

and rewrite

$$\mathsf{Q}_\mathsf{i}^\mathsf{l} = q_\mathsf{i}^{\mathsf{l},+} \cdot \mathcal{Q}_+ + i\, q_\mathsf{i}^{\mathsf{l},-} \cdot \mathcal{Q}_- + s_\mathsf{i}^{\mathsf{l},+} \cdot \mathcal{S}_+ + i\, s_\mathsf{i}^{\mathsf{l},-} \cdot \mathcal{S}_-, \tag{6.4}$$

where now the coefficients $q^\pm$ and $s^\pm$ are simply real. We will present our derivation in this basis, because it makes it straightforward to check when the reality conditions present an obstruction to the unimodular extension.

## 6.1 Unimodular extension of rank-2 solutions

One can go through Tables 1 and 2, that contain the summary of all the rank-2 bosonic solutions, and check when it is possible to construct unimodular extensions. In order to do that, we must

---

[15]Notice that this is an automorphism of the algebra only if $\hat{\xi} = 0$, but we could use this transformation even if $\hat{\xi} \neq 0$ because ultimately we are interested in the classification of the $R$-matrices rather than the algebras.

[16]This means that $q \cdot Q = q^{\alpha a}Q_{\alpha a}$, $\overline{q} \cdot \overline{Q} = \overline{q}_{\dot{\alpha} a}\overline{Q}^{\dot{\alpha} a}$, $s \cdot S = s^\alpha{}_a S_\alpha{}^a$ and $\overline{s} \cdot \overline{S} = \overline{s}_{\dot{\alpha}}{}^a \overline{S}^{\dot{\alpha}}{}_a$. As explained in appendix B, Lorentz indices $\alpha, \dot{\alpha}$ are raised and lowered with $\epsilon$ while the $\mathcal{R}$-indices $a$ are raised and lowered with the matrix $K$. Given that both matrices are antisymmetric, the contraction preserves the sign if both indices are changed of position, e.g. $q \cdot Q = q^{\alpha a}Q_{\alpha a} = q_{\alpha a}Q^{\alpha a}$. Otherwise the overall sign changes.

[17]Importantly, here the $a, b$ indices are raised and lowered with the Kronecker $\delta$ rather than the matrix $K$.

[18]This definition breaks Lorentz and $\mathcal{R}$-symmetry covariance but is useful for practical calculations in our case.

construct two odd elements

$$\mathsf{Q_i} = q_i^+ \cdot \mathcal{Q}_+ + i\, q_i^- \cdot \mathcal{Q}_- + s_i^+ \cdot \mathcal{S}_+ + i\, s_i^- \cdot \mathcal{S}_-, \tag{6.5}$$

with $\mathsf{i} = 1, 2$ that satisfy the relations in (3.10) for a given $\mathsf{h}$ and $\mathsf{e}$. If this is possible, we will denote the unimodular extension of $R$ by $\bar{R}$.

## Obstructions to the unimodularity extension

Let us present explicitly the calculations for one case that does not admit a unimodular extension, namely $\mathsf{h} = D + aJ_{03}$ and $\mathsf{e} = p_1$. In general, the best strategy is to first impose the conditions in (3.10) that are linear in $\mathsf{Q_i}$. Starting with $[\mathsf{Q_i}, \mathsf{e}] = 0$ one immediately finds that this case implies $s^\pm = 0$, so that $\mathsf{Q_i}$ can be only linear combinations of the supercharges $\mathcal{Q}_\pm$. After imposing also $[\mathsf{h}, \mathsf{Q_i}] = \frac{1}{2}\mathsf{Q_i} - \epsilon_{\mathsf{ij}} \xi \mathsf{Q_j}$, one finds that the only option that (potentially) does not break the reality of the free coefficients $a, \xi, q^\pm$ is setting just $\xi = a = 0$. At this point we turn to the relation quadratic in the odd elements, namely $\{\mathsf{Q_i}, \mathsf{Q_j}\} = -i\delta_{\mathsf{ij}}\mathsf{e}$. This is where the real basis for the supercharges turns out to be useful. Omitting the $\mathsf{i}$ index now, we find that, in order to satisfy $\{\mathsf{Q}, \mathsf{Q}\} = -i\mathsf{e}$, the following equations must hold

$$\sum_{a=1}^{4}(q_{\alpha a}^+)^2 + \sum_{a=1}^{4}(q_{\alpha a}^-)^2 = 0, \qquad \alpha = 1, 2,$$

$$\sum_{a=1}^{4}(q_{1a}^+ q_{2a}^+ + q_{1a}^- q_{2a}^-) = -\tfrac{1}{4}, \qquad \sum_{a=1}^{4}(q_{1a}^+ q_{2a}^- - q_{1a}^- q_{2a}^+) = 0. \tag{6.6}$$

As the coefficients are real, the two equations in the first line are solved only by $q_{\mathsf{i},\alpha a}^\pm = 0$. This solution is not compatible with the first equation of the second line, and in fact makes $\mathsf{Q_i}$ vanish completely. Therefore for $\mathsf{h} = D + aJ_{03}$ and $\mathsf{e} = p_1$ it is not possible to construct a unimodular extension.

The calculations are similar for all other cases of Table 2, and we will not present the details for all of them. The obstructions originate from reality conditions either from the linear commutation relations with $\mathsf{h}$ or from the quadratic anticommutation relations. As indicated in Table 1, however, it is possible to construct unimodular extensions when $\mathsf{e} = p_0$ or $\mathsf{e} = p_0 + p_3$, as we will now show.

## Allowed unimodular extensions

Let us start from those with $\mathsf{e} = p_0 + p_3$, namely from $R_1$ to $R_5$ of Table 1 included. With this choice, imposing $[\mathsf{Q_i}, \mathsf{e}] = 0$ sets $s_{\mathsf{i},1a}^\pm = 0$ and thus kills a total of 16 parameters. To proceed we need to specify the choice for $\mathsf{h}$ in each $R_i$.

### $R_1$

When taking $\mathsf{h} = (1+b)D + bJ_{03} + aJ_{12}$ and imposing $[\mathsf{h}, \mathsf{Q_i}] = \frac{1}{2}\mathsf{Q_i} - \epsilon_{\mathsf{ij}} \check{\xi} \mathsf{Q_j}$ we find that we must set $\check{\xi} = \frac{a}{2}$ and that we have six branches for the solutions, depending on whether the parameters $a$, $b$ and/or $b + 1$ are generic or vanishing.[19] Nevertheless, after the quadratic commutation relations the six branches collapse to only two branches, $a$ is generic or $a = 0$.

---

[19]As in the bosonic case, the relations in (3.12) are symmetric under $\check{\xi} \to -\check{\xi}$ and $\mathsf{Q}_1 \leftrightarrow \mathsf{Q}_2$. That means that we also have solutions with $\check{\xi} = -a/2$. We do not write these explicitly because they give rise to the same $R$-matrix and deformation as the above solution.

- $a = 0$

If $b \neq 0, -1$ we find that we must set

$$s_{i,\alpha a}^{\pm} = 0, \qquad q_{i,1a}^{\pm} = 0. \tag{6.7}$$

The non-trivial coefficients $q_{i,2a}^{\pm}$ will be further constrained by the condition $\{Q_i, Q_j\} = -i\delta_{ij}e$. In particular we find

$$\sum_a [(q_{i,2a}^{+})^2 + (q_{i,2a}^{-})^2] = \tfrac{1}{2}, \quad \text{for } i = 1, 2, \qquad \sum_a (q_{1,2a}^{-} q_{2,2a}^{-} + q_{1,2a}^{+} q_{2,2a}^{+}) = 0. \tag{6.8}$$

If $b = 0$ we find less-restrictive conditions from the linear commutation relations with $e$ and $h$, namely $s_{i,\alpha a}^{\pm} = 0$, but real solutions of the anti-commutation relations reduce this case back to (6.7) and (6.8). Similarly, if $b = -1$, the linear commutation relations set only $s_{i,1a}^{\pm} = q_{i,1a}^{\pm} = 0$ but reality of the anti-commutation relations reduce this case back to (6.7) and (6.8). We will denote this unimodular extension by $\bar{R}_1$.

- $a$ generic

If $b \neq 0, -1$ we find from the linear commutation relations with $e$ and $h$ that we must set

$$s_{i,\alpha a}^{\pm} = 0, \qquad q_{i,1a}^{\pm} = 0, \qquad q_{i,2a}^{+} = \epsilon_{ij} q_{j,2a}^{-}. \tag{6.9}$$

The remaining quadratic conditions $\{Q_i, Q_j\} = -i\delta_{ij}e$ can then be written as equations for the coefficients $q_{i,2a}^{-}$ only, reading

$$\sum_{i,a} (q_{i,2a}^{-})^2 = \tfrac{1}{2}. \tag{6.10}$$

If $b = 0$ we find again a less restrictive solution for the linear commutation relations, namely $s_{i,\alpha a}^{\pm} = 0$, $q_{2,1a}^{+} = q_{1,1a}^{-} = 0$, $q_{1,1a}^{+} = -q_{2,1a}^{-}$, $q_{i,2a}^{+} = \epsilon_{ij} q_{j,2a}^{-}$. As before, to solve the quadratic equations $\{Q_i, Q_j\} = -i\delta_{ij}e$ over the real numbers, however, we must set $q_{1,1a}^{+} = 0$ and we reduce back to the previous case (6.9) and (6.10). Similarly, if $b = -1$, solving the linear and quadratic equations over the reals reduces the conditions to (6.9) and (6.10). We will denote this unimodular extension by $\bar{R}_{1'}$.

In summary, we only have to distinguish the $a = 0$ case, enforcing (6.7) and (6.8), and the case in which $a$ is generic, enforcing (6.9) and (6.10). Both cases admit solutions over the real numbers. Note that (6.9) and (6.10) will solve (6.7) and (6.8), but not the other way around, and therefore the solutions $\bar{R}_{1'}$ reduce to a *special* solution of the $\bar{R}_1$ when taking the limit $a \to 0$.

## $R_2$

It turns out that the calculations for the case $h = \tfrac{1}{2}(D - J_{03}) + aJ_{12} + \alpha(p_0 - p_3)$ with $\alpha^2 = 1$ are completely analogous to the previous one. In fact, the parameter $b$ of $R_1$ plays no role in the classification of the solutions, and the extra contribution with $p_0 - p_3$ is harmless because the supercharges $Q, \overline{Q}$ commute with the momenta $p_i$. If $a = 0$, the solutions are then given by (6.7) and (6.8), whose corresponding $R$-matrix we denote by $\bar{R}_2$, or if $a$ is generic they are given by (6.9) and (6.10), denoted by $\bar{R}_{2'}$.

## $R_3$

The calculations for $h = -J_{03} + \alpha p_1$ with $\alpha^2 = 1$ are analogous to the case $R_1$ with the condition $a = 0$, so that the solutions are given by (6.7) and (6.8), whose $R$-matrix will be denoted by $\bar{R}_3$.

### $R_4$

The calculations for $\mathsf{h} = D + \alpha(J_{01} - J_{13})$ with $\alpha^2 = 1$ are also analogous to the case $R_1$ with the condition $a = 0$. In fact, the element $J_{01} - J_{13}$ has non-vanishing commutators with supercharges with $\alpha = 1$, but these contributions are already set to zero by other conditions. Therefore, the solutions are given by (6.7) and (6.8), whose $R$-matrix will be denoted by $\bar{R}_4$.

### $R_5$

Here we have $\mathsf{h} = \frac{1}{2}(D - J_{03}) + aJ_{12} + b(k_0 + k_3 + 2p_3)$ and $\mathsf{e} = p_0 + p_3$, which compared to $R_1$ is a genuinely new case only if $b \neq 0$. The results are in fact identical to $R_1$, and therefore if $a = 0$ they are given by (6.7) and (6.8), whose $R$-matrix we denote by $\bar{R}_5$, and if $a$ is generic they are given by (6.9) and (6.10), with $R$-matrix $\bar{R}_{5'}$.

### $R_6$

Let us now consider the choice $\mathsf{h} = D + aJ_{12}$ and $\mathsf{e} = p_0$. Imposing $[\mathsf{Q}_i, \mathsf{e}] = 0$ we find that all $s^{\pm} = 0$. After imposing $[\mathsf{h}, \mathsf{Q}_i] = \frac{1}{2}\mathsf{Q}_i - \epsilon_{ij}\,\check{\xi}\,\mathsf{Q}_j$ we conclude that we must impose

$$\check{\xi} = \frac{a}{2}, \qquad \text{and} \qquad q_{i,\alpha a}^+ = (-1)^\alpha \epsilon_{ij} q_{j,\alpha a}^-. \tag{6.11}$$

Notice that we are able to solve for all the $q^+$ coefficients, for example, so that the remaining constraints will be imposed only on the coefficients $q_{i,\alpha a}^-$. In this case the condition $\{\mathsf{Q}_i, \mathsf{Q}_j\} = -i\delta_{ij}\mathsf{e}$ is equivalent to the following equations

$$\sum_{i,a}(-1)^i\, q_{i,1a}^- q_{i,2a}^- = 0, \qquad \sum_{a=1}^4 (q_{1,1a}^- q_{2,2a}^- + q_{1,2a}^- q_{2,1a}^-) = 0,$$
$$\sum_{i,a}(q_{i,\alpha a}^-)^2 = \tfrac{1}{4}, \qquad \alpha = 1, 2. \tag{6.12}$$

The above equations admit solutions over the real numbers, and therefore the choice $\mathsf{h} = D + aJ_{12}$ and $\mathsf{e} = p_0$ admits unimodular extensions, denoted as $\bar{R}_6$.

In section 6.3 we will discuss how to reduce the number of parameters appearing in $\mathsf{Q}_1, \mathsf{Q}_2$ by exploiting the inner $\mathfrak{so}(6)$ automorphisms of the algebra and possibly also the residual $\mathfrak{so}(2,4)$ automorphisms. This discussion will make it easier to identify the allowed solutions to the above quadratic equations that characterise the unimodular extensions.

## 6.2 Unimodular extension of the rank-4 and rank-6 cases

All the rank-4 and rank-6 cases that do not have[20] $J_{12}$ in $\mathsf{h}$ admit the generalisation of the solution (6.7) and (6.8) found for rank-2. In particular we have

$$\check{\xi} = 0, \qquad s_{i,\alpha a}^{\pm,\mathsf{I}} = 0, \qquad q_{i,1a}^{\pm,\mathsf{I}} = 0. \tag{6.13}$$

---

[20]That is, all the extended solutions of section 2, with the extra assumption of setting $a = 0$ in $R_{12}$. Notice, however, that (up to automorphisms) when $a = 0$ $R_{12}$ reduces to a special case of $R_7$ and therefore cannot be considered genuinely new.

and the conditions $\{Q_i^I, Q_j^J\} = -i\delta^{IJ}\delta_{ij}e$ read

$$\sum_a [(q_{i,2a}^{+,I})^2 + (q_{i,2a}^{-,I})^2] = \tfrac{1}{2}, \quad \text{for } i = 1, 2, \ I = 1, \ldots, N_1,$$

$$\sum_a (q_{i,2a}^{-,I} q_{j,2a}^{-,J} + q_{i,2a}^{+,I} q_{j,2a}^{+,J}) = 0, \quad \text{with } (I, i) \neq (J, j). \tag{6.14}$$

The corresponding $R$-matrices will be denoted by $\bar{R}_7$–$\bar{R}_{11}$ and $\bar{R}_{13}$, $\bar{R}_{14}$.

For a non-trivial extension of the $R_{12}$ solution, denoted by $\bar{R}_{12}$, we must take the generalisation of the solution (6.9) and (6.10). We must assume $a \neq 0$ and set

$$s_{i,\alpha a}^{\pm,I} = 0, \qquad q_{i,1a}^{\pm,I} = 0, \qquad q_{i,2a}^{+,I} = \epsilon_{ij} q_{j,2a}^{-,I}. \tag{6.15}$$

Now the extra conditions from $\{Q_i^I, Q_j^J\} = -i\delta^{IJ}\delta_{ij}e$ are equivalent to

$$\sum_{i,a} (q_{i,2a}^{-,I})^2 = \tfrac{1}{2}, \quad \text{for } I = 1, \ldots, N_1,$$

$$\sum_{i,a} q_{i,2a}^{-,1} q_{i,2a}^{-,2} = 0, \qquad \sum_{i,j,a} \epsilon^{ij} q_{i,2a}^{-,1} q_{j,2a}^{-,2} = 0. \tag{6.16}$$

## 6.3 Simplifications by inner automorphisms

The supercharges found in the previous sections still have a large number of free parameters that in fact are not all physical. They can be removed by exploiting inner $\mathfrak{so}(6)$ automorphisms, as well as in principle residual inner automorphisms of $\mathfrak{so}(2,4)$ that leave $h$ and $e$ invariant. The latter are summarised for each rank-2 bosonic $R$-matrix admitting a unimodular extension in Table 1.

Before analysing each $R$-matrix $\bar{R}_i$ separately, let us first argue in general how the simplification by $SO(6)$ can be performed. Consider a general supercharge $Q_i$ of $\mathfrak{psu}(2,2|4)$ of the form (6.4). We know already that all unimodular extensions require $s_{i,\alpha a}^{\pm,I} = 0$ and thus we start from

$$Q_i = q_i^+ \cdot \mathcal{Q}_+ + i q_i^- \cdot \mathcal{Q}_- , \tag{6.17}$$

where for now we will suppress the index $I = 1, \ldots, N_1$. There are 15 generators of $\mathfrak{so}(6)$ that we can now exploit to simplify $Q_i$. As follows from (B.13), their corresponding adjoint actions are dictated by

$$[\mathcal{R}_{AB}, (\mathcal{Q}_\pm)^{\alpha a}] = \frac{1}{2}(\rho_{AB}^{\text{AS}})^a{}_b (\mathcal{Q}^\pm)^{\alpha b} - \frac{1}{2}(\rho_{AB}^{\text{S}})^a{}_b (\mathcal{Q}^\mp)^{\alpha b}, \tag{6.18}$$

with $\rho_{AB}^{\text{S}}$ ($\rho_{AB}^{\text{AS}}$) the symmetric (antisymmetric) part of $\rho_{AB}$ as matrices in the $a, b$ indices. In our anti-hermitian matrix realisation of $\mathfrak{so}(6)$ (see appendix B), there are precisely $\binom{4}{2} = 6$ generators for which $\rho_{AB}$ is purely antisymmetric in the $a, b$ indices and that will act as a simple rotation on the $a = 1, 2, 3, 4$ of $q_{i,\alpha a}^\pm$. Let us call $\mathcal{R}^{(r)}$ the set of these generators. The other 9 generators are purely imaginary with symmetric $\rho_{AB}$ and will thus mix $q_{i,\alpha a}^+$ and $q_{i,\alpha a}^-$ with $i, \alpha$ fixed. They can be divided into 3 generators corresponding to phase transformations rotating $q_{i,\alpha a}^+$ and $q_{i,\alpha a}^-$ with also $a$ fixed, whose set we will call $\mathcal{R}^{(p)}$, and $\binom{4}{2} = 6$ generators corresponding to rotations on the $a$ indices between $q_{i,\alpha a}^+$ and $q_{i,\alpha a}^-$, whose set we will call $\mathcal{R}^{(i)}$. For clarity, we can depict this as

$$\mathcal{R}^{(r)}: \quad q_{i,\alpha a}^\pm \leftrightarrow q_{i,\alpha b}^\pm, \quad \text{with} \quad (a, b) \in \{(1, 2), (1, 3), (1, 4), (2, 3), (2, 4), (3, 4)\},$$

$$\mathcal{R}^{(i)}: \quad q_{i,\alpha a}^\pm \leftrightarrow q_{i,\alpha b}^\mp, \quad \text{with} \quad (a, b) \in \{(1, 2), (1, 3), (1, 4), (2, 3), (2, 4), (3, 4)\}, \tag{6.19}$$

$$\mathcal{R}^{(p)}: \quad q_{i,\alpha a}^\pm \leftrightarrow q_{i,\alpha a}^\mp, \quad \text{with} \quad a \text{ fixed},$$

and i, $\alpha$ fixed in all cases.

Let us now illustrate how these can be used to eliminate some parameters $q_{i,\alpha a}^{\pm}$. We can start focusing, for example, on $q_{1,1a}^{\pm}$ and use the $\{(1,2),(1,3),(1,4)\}$ rotations of $\mathcal{R}^{(r)}$ and $\mathcal{R}^{(i)}$ to set $q_{1,1\{2,3,4\}}^{\pm} = 0$ and keep for $(i,\alpha) = (1,1)$ only $q_{1,11}^{\pm}$ non-trivial. Notice that the actions of $\mathcal{R}^{(r)}$ and $\mathcal{R}^{(i)}$ interfere non-trivially with each other so that on each $(a,b)$ plane the two kinds of rotations must be implemented simultaneously in order to reach the wanted gauge. The other rotations of $\mathcal{R}^{(r,i)}$ leave this choice invariant and subsequently the $\{(2,3),(2,4)\}$ can be used to set, for example, $q_{1,2\{3,4\}}^{\pm} = 0$ keeping for $(i,\alpha) = (1,2)$ only $q_{1,21}^{\pm}$ and $q_{1,22}^{\pm}$ non-trivial. Within $\mathcal{R}^{(r,i)}$ we are finally left with $(3,4)$, again leaving our previous choices invariant, and which can be used to set, for example, $q_{2,14}^{\pm} = 0$. At last there are also the $\mathcal{R}^{(p)}$ to exploit. Importantly, so far we have treated the superscripts $+$ and $-$ in the same way (when keeping the remaining indices i, $\alpha$, a fixed). This is crucial at the moment of using the $\mathcal{R}^{(p)}$ transformations, because it means that the previous choices for vanishing coefficients are not spoiled. Considering for example the basis

$$\mathcal{R}^{(p)} : \quad \rho^a{}_b \in \{\mathrm{diag}(i,0,0,-i), \mathrm{diag}(0,i,0,-i), \mathrm{diag}(0,0,i,-i)\}, \tag{6.20}$$

we can use the first two elements to set, for example, $q_{1,11}^{+} = q_{1,22}^{+} = 0$, and the last element (leaving the latter choice invariant) to set, for example, $q_{2,13}^{+} = 0$.

The most convenient "gauge" choices to be made will depend on the specific cases under study, to which we now turn. In particular, the type of constraints following from the (anti)-commutation relations required for the unimodular extensions will play an important role in this. Note that these constraints will of course be left invariant after the action of inner automorphisms that leave h and e invariant. If possible, we may also employ the residual inner $\mathfrak{so}(2,4)$ automorphisms of each case. We first consider the rank-2 extensions and comment on the possible higher-ranks at the end.

## $\bar{R}_1$ and $\bar{R}_{1'}$

Given the results of section 6.1, we could distinguish here two cases, i.e. $a$ generic or $a = 0$. In both cases, only the parameters $q_{i,2a}^{\pm}$ were left non-vanishing after imposing the required (anti)commutation relations. With the exception of $J_{12}$ (which acts as a phase transformation $q_{i,\alpha a}^{\pm} \leftrightarrow q_{i,\alpha a}^{\mp}$ with fixed indices i, $\alpha$ and $a$), the inner automorphisms of Table 1 then all act trivially on the supercharges and therefore their possible usage can be discarded.

- $\bar{R}_1$. In this case, the parameters $q_{i,2a}^{\pm}$ are subject to the constraints (6.8). As illustrated above, we can use $\{(1,2),(1,3),(1,4)\}$ of $\mathcal{R}^{(r,i)}$ to keep only $q_{1,21}^{\pm}$ and $q_{2,2a}^{\pm}$. Next, we can use $\{(2,3),(2,4)\}$ of $\mathcal{R}^{(r,i)}$ to keep only $q_{1,21}^{\pm}$, $q_{2,21}^{\pm}$, and $q_{2,22}^{\pm}$. At last, we can use the first two elements of (6.20) to reduce this to $q_{1,21}^{+}$, $q_{2,21}^{\pm}$, and $q_{2,22}^{\pm}$. The constraints (6.8) are then solved by $q_{2,21}^{+} = 0$ and[21]

$$q_{1,21}^{+} = \frac{1}{\sqrt{2}}, \qquad (q_{2,22}^{+})^2 + (q_{2,21}^{-})^2 = \frac{1}{2} . \tag{6.21}$$

Note that, the remaining $(3,4)$ of $\mathcal{R}^{(r,i)}$ and the last element of (6.20) do not act on these parameters. These transformations can be used to simplify higher-rank extensions, as we will do below starting from $R_7$. Similarly, one can check that the residual $J_{12}$ action on our reduced

---

[21]The quadratic equation admits also the solution for $q_{1,21}^{+}$ with the negative sign. A change of sign on $\mathsf{Q}_1$ or $\mathsf{Q}_2$ however leaves the $R$-matrix (3.11) invariant.

parameter space does not simplify this case further, in fact the action of $J_{12}$ is not independent from that of the first two elements of (6.20).

- $\bar{R}_{1'}$. In this case, the parameters $q^{\pm}_{i,2a}$ are subjected to different constraints, namely $q^+_{i,2a} = \epsilon_{ij} q^-_{j,2a}$ and (6.10). As before, we can now exploit the $\{(1,2),(1,3),(1,4)\}$ of $\mathcal{R}^{(r,i)}$ to set $q^{\pm}_{1,2\{2,3,4\}} = 0$. The constraint $q^+_{i,2a} = \epsilon_{ij} q^-_{j,2a}$ then actually implies that only the 4 coefficients $q^{\pm}_{i,21}$ are left non-trivial and they are related as $q^+_{1,21} = q^-_{2,21}, q^+_{2,21} = -q^-_{1,21}$. After this we still have the freedom $\{(2,3),(2,4),(3,4)\}$ of $\mathcal{R}^{(r,i)}$. In this case, they are not useful because they rotate among each other coefficients that are already vanishing. One could imagine using them to eliminate 6 more parameters of a higher-rank unimodular extension, however this case (i.e. the parameter $a$ generic) does not admit such extensions. The first element of (6.20) can subsequently be used to set, for example, $q^+_{2,21} = 0$ which upon the constraints also implies $q^-_{1,21} = 0$. We are left with one parameter which must be fixed by solving (6.10). We take[22]

$$q^-_{2,21} = q^+_{1,21} = \frac{1}{\sqrt{2}}\ . \tag{6.22}$$

Note that here we did not need the residual $J_{12}$ automorphisms. In fact, it does not leave the choice $q^+_{1,21} = q^-_{2,21} = 0$ invariant.

Let us remark that in the special case for $\bar{R}_1$ with $q^+_{2,22} = 0$, the solution (6.21) reduces to (6.22) and thus the latter holds for truly generic $a$, as it should.

## $\bar{R}_2$, $\bar{R}_{2'}$, $\bar{R}_3$, $\bar{R}_4$, $\bar{R}_5$, and $\bar{R}_{5'}$

All these cases are analogous to the case $\bar{R}_1$ and $\bar{R}_{1'}$. The $\bar{R}_2$, $\bar{R}_3$, $\bar{R}_4$, and $\bar{R}_5$ are solved by (6.21), while $\bar{R}_{2'}$ and $\bar{R}_{5'}$ are solved by (6.22), with all the other parameters set to zero. Note, however, that certain versions of the $\bar{R}_{5'}$ have a corresponding higher-rank extension, i.e. $\bar{R}_{12}$, for which we will be able to use the remaining $SO(6)$ transformations that have not been exploited in the $\bar{R}_{1'}$ case (see below).

## $\bar{R}_6$

In this case, all the $q^{\pm}_{i,\alpha a}$ were left non-vanishing, and they were subjected to $q^+_{i,\alpha a} = (-1)^\alpha \epsilon_{ij} q^-_{j,\alpha a}$ and (6.12). Following now precisely the illustration of the beginning of this section, we can use all of the $\mathfrak{so}(6)$ elements to set $q^{\pm}_{1,1\{2,3,4\}} = q^{\pm}_{1,2\{3,4\}} = q^{\pm}_{2,14} = q^+_{1,11} = q^+_{1,22} = q^+_{2,13} = 0$. The constraint $q^+_{i,\alpha a} = (-1)^\alpha \epsilon_{ij} q^-_{j,\alpha a}$ then further implies that the only non-vanishing parameters are $q^{\pm}_{1,21} = \pm q^{\mp}_{2,21}$, $q^-_{1,11} = q^+_{2,11}$, and $q^-_{1,22} = -q^+_{2,22}$. This, in fact, suffices to completely solve (6.12). We find that we must set $q^{\pm}_{1,21} = 0 = q^{\mp}_{2,21}$ and[23]

$$q^-_{1,11} = q^+_{2,11} = q^-_{1,22} = -q^+_{2,22} = \frac{1}{2}\ . \tag{6.23}$$

---

[22] Again the choice of sign on $q^-_{2,21}$ would not affect the resulting $R$-matrix.

[23] When solving the quadratic equations we actually find $q^-_{1,11} = q^+_{2,11} = \pm\frac{1}{2}, q^-_{1,22} = -q^+_{2,22} = \pm\frac{1}{2}$ where the two choices of signs are uncorrelated. However, we first notice that the 4 choices of combinations of signs lead to only 2 independent solutions for the $R$-matrix, namely when the signs either agree or are opposite. This is due to the fact that changing the overall sign to $\mathsf{Q}_1$ or $\mathsf{Q}_2$ does not change the $R$-matrix. Nevertheless, we find that these two seemingly distinct choices (consider $\mathsf{Q}_i$ with $q^-_{1,11} = q^-_{1,22} = 1/2$ and $\mathsf{Q}'_i$ with $q^-_{1,11} = -q^-_{1,22} = 1/2$) are related by an inner automorphism $\mathsf{Q}_i = M\mathsf{Q}'_i M^{-1}$ with $M \in PSU(2,2|4)$. In fact several elements $M \in PSU(2,2|4)$ realise this, including first swapping $\mathsf{Q}_1 \leftrightarrow \mathsf{Q}_2$ and then acting with $J_{12}$ (which effectively exchanges the superscripts $+$ and $-$).

All the parameters are determined, and we did not need to use the residual $\mathfrak{so}(2,4)$ automorphisms. Interestingly, note that even though we exploited the $(3,4)$ of $\mathcal{R}^{(r,i)}$ and the latter element of (6.20), they are restored as inner automorphisms in this case because of the constraint $q^+_{i,\alpha a} = (-1)^\alpha \epsilon_{ij} q^-_{j,\alpha a}$.

## $\bar{R}_7, \bar{R}_8, \bar{R}_9, \bar{R}_{10},$ and $\bar{R}_{11}$

These $R$-matrices originate from bosonic rank-4 $R$-matrices and they thus have the $Q^l_i$ with $i = 1, 2$ and $l = 1, 2$. Their only non-vanishing parameters after the required commutation relations are $q^{\pm,l}_{i,2a}$ subjected to the constraints (6.14). As we know from section 6.2, these cases are similar to the unimodular extension $\bar{R}_1$, and in fact using $\{(1,2),(1,3),(1,4),(2,3),(2,4)\}$ of $\mathcal{R}^{(r,i)}$ and the first two elements of (6.20) as before, the equations (6.14) for $q^{\pm,1}_{i,2a}$ can be solved as in (6.21), i.e. $q^{+,1}_{2,21} = 0$ and

$$q^{+,1}_{1,21} = \frac{1}{\sqrt{2}}, \qquad (q^{+,1}_{2,22})^2 + (q^{-,1}_{2,21})^2 = \frac{1}{2} \ . \tag{6.24}$$

Now we can focus on $q^{\pm,2}_{i,2a}$ and use the remaining $\mathfrak{so}(6)$ elements to simplify them. First, with $\{(3,4)\}$ of $\mathcal{R}^{(r,i)}$ we can set $q^{\pm,2}_{1,24} = 0$. The last element of (6.20) can be used to set $q^{-,2}_{1,23} = 0$. Note that we do not have further inner automorphisms of $\mathfrak{so}(2,4)$ that leave $\{\mathsf{h},\mathsf{e},\mathsf{e}_+,\mathsf{e}_-\}$ invariant and that would act non-trivially on the supercharges. One can now check that (6.14) further enforces $q^{+,2}_{i,21} = 0$ and leaves us with 13 parameters subjected to 6 constraints. We checked that these can admit real solutions with 7 remaining free parameters in the supercharges $Q^l_i$. We will not analyse them further here.

## $\bar{R}_{12}$

Recall that for a genuinely new unimodular extension $\bar{R}_{12}$ we must assume $a \neq 0$ and thus this will be similar to the $\bar{R}_{1'}$ case. After the required (anti-)commutation relations, the non-vanishing parameters are $q^{\pm,l}_{i,2a}$ subjected to $q^{+,l}_{i,2a} = \epsilon_{ij} q^{-,l}_{j,2a}$ and (6.16). We exploit the $\{(1,2),(1,3),(1,4)\}$ of $\mathcal{R}^{(r,i)}$ as well as the first element of (6.20) to set $q^{\pm,1}_{1,2\{2,3,4\}} = 0$ and $q^{-,1}_{2,21} = 0$. The constraints for $l = 1$ then completely fix $Q^1_i$ and in particular imply that also $q^{\pm,1}_{2,2\{2,3,4\}} = 0$ and $q^{+,1}_{1,21} = 0$ as well as

$$q^{-,1}_{1,21} = -q^{+,1}_{2,21} = \frac{1}{\sqrt{2}} \ , \tag{6.25}$$

where we made an inconsequential choice of the sign that will not affect the $R$-matrix. Now we consider $l = 2$ and possibly exploit the remaining $SO(6)$ transformations, i.e. $\{(2,3),(2,4),(3,4)\}$ of $\mathcal{R}^{(r,i)}$ as well as the latter two elements of (6.20). This allows to set respectively $q^{\pm,2}_{1,2\{3,4\}} = q^{\pm,2}_{2,24} = 0$ and $q^{-,2}_{2,22} = 0$ which implies using $q^{+,l}_{i,2a} = \epsilon_{ij} q^{-,l}_{j,2a}$ that also $q^{\pm,2}_{2,23} = 0$ and $q^{+,2}_{1,22} = 0$.[24] Hence at this stage only $q^{-,2}_{1,21} = -q^{+,2}_{2,21}$, $q^{-,2}_{2,21} = q^{+,2}_{1,21}$, and $q^{-,2}_{1,22} = -q^{+,2}_{2,22}$ are undetermined. Using the results of $Q^1_i$, the remaining constraints simply imply $q^{-,2}_{1,21} = q^{-,2}_{2,21} = q^{+,2}_{2,21} = q^{+,2}_{1,21} = 0$ and

$$q^{-,2}_{1,22} = -q^{+,2}_{2,22} = \frac{1}{\sqrt{2}} \ , \tag{6.26}$$

---

[24]In fact, we did not need to use the latter element of (6.20) which remains as a freedom that does not act on the surviving parameters.

where again we made a choice for the sign. Concluding, for the unimodular extension $\bar{R}_{12}$ we do not have any remaining free parameters in the supercharges.

### $\bar{R}_{13}$ and $\bar{R}_{14}$

These $R$-matrices originate the bosonic rank-6 $R$-matrices and are similar to the unimodular extension $\bar{R}_1$, albeit now with supercharges $Q_i^l$ with $i = 1, 2$ and $l = 1, 2, 3$. While the $\bar{R}_1$ (from rank-2) had one remaining free parameter, their rank-4 generalisations admit solutions with 7 remaining free parameters (see $\bar{R}_7$–$\bar{R}_{11}$ discussed above) after using residual inner automorphisms. The rank-6 generalisation $\bar{R}_{13}$ and $\bar{R}_{14}$ (where no further inner automorphisms can be exploited) will have in principle 16 more initial parameters than the rank-4 cases which all have to satisfy the same 10 constraints as before, i.e. (6.14). Because of these reasons, it is clear that the $\bar{R}_{13}$ and $\bar{R}_{14}$ have a complicated and large system to solve with a large number of remaining free parameters. We will not analyse this further.

## 7 Preserved isometries and superisometries

In this section, we consider the unimodular extensions of the rank-2 $R$-matrices and identify the generators of $T_{\bar{A}} \in \mathfrak{psu}(2,2|4)$ whose adjoint action commutes with the action of $\bar{R}$, i.e.

$$\mathrm{ad}_{T_{\bar{A}}} \bar{R} = \bar{R} \, \mathrm{ad}_{T_{\bar{A}}} \, . \tag{7.1}$$

Because of the Jacobi identity, these generators span a subalgebra of $\mathfrak{psu}(2,2|4)$. They have the important interpretation that they correspond to (super)isometries of the supergravity background. Indeed, at the group level, they generate global left transformations $g \to g_L g$ with $g_L \in PSU(2,2|4)$ constant and satisfying

$$\mathrm{Ad}_{g_L}^{-1} \bar{R} \, \mathrm{Ad}_{g_L} = \bar{R} \, , \tag{7.2}$$

which leaves the deformed semisymmetric coset sigma-model action (A.6) invariant.

We collect our results in Table 6, which is the complete version of Table 3 of our summary section. Because the $R$-matrix preserves the degree of the superalgebra element, we can consider bosonic and fermionic generators $T_{\bar{A}} \in \mathfrak{psu}(2,2|4)$ separately. For the fermionic generators, we only list the number of independent generators, corresponding to the number of superisometries preserved. We also use a short-hand notation for those elements $T_{\bar{A}}$ that are strictly in $\mathfrak{so}(6)$, as they are repeated frequently. We define a 9-dimensional algebra,

$$\mathfrak{k}_1 \equiv \mathrm{span}(R_{16} - R_{24}, R_{14} + R_{26}, R_{36} + R_{45}, R_{34} + R_{56}, R_{13} + R_{25}, R_{15} - R_{23}, R_{12}, R_{35}, R_{46}) \, , \tag{7.3}$$

which is the algebra appearing once the supercharges $Q_i$ are completely determined, with the exception of the distinct $\bar{R}_6$ case. The elements of $\mathfrak{k}_1$ correspond to the remaining $\mathfrak{so}(6)$ automorphisms that were not exploited in the previous section, i.e. the $\{(2,3),(2,4),(3,4)\}$ of $\mathcal{R}^{(r,i)}$ as well as the last two phase transformations of (6.20), in addition to the first phase transformation of (6.20).[25] It is not hard to show that the algebra $\mathfrak{k}_1$ is isomorphic to $\mathfrak{su}(3) \oplus \mathfrak{u}(1)$ by identifying a central element and calculating the dual Coxeter number of the remaining 8-dimensional algebra. We also define the following two 4-dimensional algebras

$$\mathfrak{k}_2 \equiv \mathrm{span}(R_{13} + R_{25}, R_{15} - R_{23}, R_{12} - R_{35}, R_{13} + \alpha R_{35}) \simeq \mathfrak{su}(2) \oplus \mathfrak{u}(1), \tag{7.4}$$

$$\mathfrak{k}_3 \equiv \mathrm{span}(R_{13} + R_{25}, R_{15} - R_{23}, R_{12} - R_{35}, R_{12} + R_{35}) \simeq \mathfrak{su}(2) \oplus \mathfrak{u}(1), \tag{7.5}$$

---

[25]Note that the requirement of an inner automorphism that leaves $\{h, e, Q_i\}$ invariant, i.e. $[x, h] = [x, e] = [x, Q_i] = 0$, is stronger than the requirement of a (super)isometry.

where $\alpha = q_{2,21}^-/q_{2,22}^+$ with the assumption $q_{2,22}^+ \neq 0$. The algebra $\mathfrak{k}_2$ appears for those cases in which the supercharges $Q_i$ have a free parameter, while $\mathfrak{k}_3$ is the subalgebra of isometries of $\mathfrak{so}(6)$ for $\bar{R}_6$. The first three elements of $\mathfrak{k}_2$ and $\mathfrak{k}_3$ correspond to the $(3,4)$ of $\mathcal{R}^{(r,i)}$ and the latter element of $(6.20)$, which are all residual $\mathfrak{so}(6)$ inners of the respective cases. Furthermore, we remark that even though the algebra $\mathfrak{k}_2$ appears to depend on a continuous parameter, its structure constants are $\alpha$-independent once we redefine $R_{13} + \alpha R_{35} \to R_{13} + \alpha R_{35} + \frac{\alpha}{2}(R_{12} - R_{35})$. A further appropriate shift of this element with $R_{13} + R_{25}$ exposes that $\mathfrak{k}_2 \simeq \mathfrak{su}(2) \oplus \mathfrak{u}(1)$. How this algebra is embedded in $\mathfrak{so}(6)$, however, does depend on $\alpha$.

From the point of view of the sigma-model action, the extra elements in $\mathfrak{k}_1, \mathfrak{k}_2, \mathfrak{k}_3$ on top of the residual $\mathfrak{so}(6)$ inners seem to imply the possibility of further simplifications of the supercharges. Nevertheless, these transformations do not act independently on the reduced parameter space or compared to the $J_{12}$ action.

We will not analyse the preserved (super)isometries of the unimodular extensions of the higher-rank solutions as in these cases, with exception of $\bar{R}_{12}$, the supercharges have a large number of remaining free parameters which highly complicates their study. For $\bar{R}_{12}$, however, all the parameters in $Q_i$ are completely determined, and we find for any value of $a \neq 0$ that there are no residual superisometries.

| $\bar{R}$ | conditions | $T_{\bar{A}} \in \mathfrak{so}(2,4) \oplus \mathfrak{so}(6)$ | supercharges |
|---|---|---|---|
| 1 | $a = 0$ | $D + J_{03}, p_0 + p_3, J_{12} - R_{46}, \mathfrak{k}_2$ | 0 |
| | $a = 0, b = -1$ | $D + J_{03}, p_0 + p_3, p_1, p_2, J_{12} - R_{46}, \mathfrak{k}_2$ | 0 |
| | $a = 0, b = 0$ | $D + J_{03}, p_0 + p_3, J_{01} - J_{13}, J_{02} - J_{23}, J_{12} - R_{46}, \mathfrak{k}_2$ | 0 |
| | $a = 0, b = -1/2$ | $D + J_{03}, p_0, p_3, k_0 + k_3, J_{12} - R_{46}, \mathfrak{k}_2$ | 8 |
| | $a = 0, q_{2,22}^+ = 0$ | $D + J_{03}, p_0 + p_3, J_{12}, \mathfrak{k}_1$ | 0 |
| | $a = 0, b = -1, q_{2,22}^+ = 0$ | $D + J_{03}, p_0 + p_3, p_1, p_2, J_{12}, \mathfrak{k}_1$ | 0 |
| | $a = 0, b = 0, q_{2,22}^+ = 0$ | $D + J_{03}, p_0 + p_3, J_{01} - J_{13}, J_{02} - J_{23}, J_{12}, \mathfrak{k}_1$ | 0 |
| | $a = 0, b = -1/2, q_{2,22}^+ = 0$ | $D + J_{03}, p_0, p_3, k_0 + k_3, J_{12}, \mathfrak{k}_1$ | 12 |
| 1' | $a \neq 0$ | $D + J_{03}, p_0 + p_3, J_{12}, \mathfrak{k}_1$ | 0 |
| | $a \neq 0, b = -1/2$ | $D + J_{03}, p_0, p_3, k_0 + k_3, J_{12}, \mathfrak{k}_1$ | 0 |
| 2 | $a = 0$ | $p_0, p_3, J_{12} - R_{46}, \mathfrak{k}_2$ | 4 |
| | $a = 0, q_{2,22}^+ = 0$ | $p_0, p_3, J_{12}, \mathfrak{k}_1$ | 6 |
| 2' | $a \neq 0$ | $p_0, p_3, J_{12}, \mathfrak{k}_1$ | 0 |
| 3 | $-$ | $p_0 + p_3, p_1, p_2, \mathfrak{k}_2$ | 0 |
| | $q_{2,22}^+ = 0$ | $p_0 + p_3, p_1, p_2, \mathfrak{k}_1$ | 0 |
| 4 | $-$ | $p_0 + p_3, J_{01} - J_{13}, J_{02} - J_{23}, \mathfrak{k}_2$ | 0 |
| | $q_{2,22}^+ = 0$ | $p_0 + p_3, J_{01} - J_{13}, J_{02} - J_{23}, \mathfrak{k}_1$ | 0 |
| 5 | $a = 0$ | $p_0 + p_3, k_0 + k_3 + 2p_3, J_{12} - R_{46}, \mathfrak{k}_2$ | 0 |
| | $a = 0, q_{2,22}^+ = 0$ | $p_0 + p_3, k_0 + k_3 + 2p_3, J_{12}, \mathfrak{k}_1$ | 0 |
| 5' | $a \neq 0$ | $p_0 + p_3, k_0 + k_3 + 2p_3, J_{12}, \mathfrak{k}_1$ | 0 |
| 6 | $-$ | $p_0, J_{12}, \mathfrak{k}_3$ | 0 |

Table 6: The bosonic generators $T_{\bar{A}} \in \mathfrak{so}(2,4) \oplus \mathfrak{so}(6)$, as well as the number of fermionic elements in $\mathfrak{psu}(2,2|4)$ that satisfy $(7.1)$ for the unimodular extended rank-2 Jordanian $R$-matrices of the form $r = \mathsf{h} \wedge \mathsf{e} - \frac{i}{2}(Q_1 \wedge Q_1 + Q_2 \wedge Q_2)$. Such generators represent the residual (super)isometries of the deformed supergravity background. If a parameter is not specified, it is assumed to be generic (modulo constraints such as $(6.21)$). The algebras $\mathfrak{k}_1, \mathfrak{k}_2,$ and $\mathfrak{k}_3$ are subalgebras of $\mathfrak{so}(6)$ and defined in $(7.3), (7.4),$ and $(7.5)$ respectively.

# 8    Conclusions

We have classified all antisymmetric bosonic Jordanian solutions of the classical Yang-Baxter equation on $\mathfrak{psu}(2,2|4)$ and constructed their most generic fermionic extensions that ensure unimodularity. These properties are siginificant for constructing a Yang-Baxter deformation of the integrable string sigma-model on $\mathfrak{psu}(2,2|4)$ which gives rise to the maximally supersymmetric $AdS_5 \times S^5$ background. In particular, antisymmetric solutions of the CYBE ensure that the deformation preserves the property of integrability. Unimodularity, in addition, ensures that the deformed $AdS_5 \times S^5$ background will still solve the type IIB supergravity equations of motion.

When the bosonic Jordanian $R$-matrices are not extended with fermionic supercharges, they are non-unimodular and the corresponding background solves the modified (or generalised) supergravity equations. We find that they are at most of rank-6, where the rank denotes the number of bosonic elements in the construction of the $R$-matrix. For all these cases, we analyse whether or not they admit a unimodular extension, which is a stringent requirement for rank-2, but is always possible for rank-4 and rank-6. For the simplest unimodular extensions, namely those of rank-2, we also analyse the preserved (super)isometries of the corresponding deformed supergravity background and find that they preserve at most 12 superisometries. All of our main results are structurally summarised in section 2.

Each of the Jordanian solutions that we have constructed are inequivalent and correspond to a different deformed supergravity background. Our results may therefore offer a wide range of applications for deformations of the $AdS_5$/SYM holographic duality. See [41–44] for some preliminary proposals of deformations of the dual gauge theory. In this paper, we were primarily concerned with the algebraic classification of (unimodular) Jordanian solutions. For the purpose of deformed holography, one may in a first stage also analyse if the dilaton is well-behaved for the simpler unimodular $R$-matrices of table 6. In this table, the case $\bar{R}_1$ ($a = 0$, $b = -1/2$, $q_{2,22}^+ = 0$) is the only one with 12 residual superisometries. It is in fact this example (up to inner automorphisms) which was first constructed in [26] (and indeed has a well-behaved dilaton) and of which the semi-classical spectral problem of the deformed sigma-model was later analysed in [36] by means of algebraic curve techniques. It would be interesting to extend this study to examples with lower residual superisometries and analyse if any obstructions occur.

As we have found, the family of Jordanian deformations of $AdS_5 \times S^5$ is quite large. This is a confirmation of a large landscape of integrable deformations of the string sigma-model, that undoubtedly extends beyond the Jordanian class.

## Acknowledgements

We thank Leander Wyss for useful discussions and collaboration during the initial stages of this project. We are supported by the fellowship of "la Caixa Foundation" (ID 100010434) with code LCF/BQ/PI19/11690019, by AEI-Spain (under project PID2020-114157GB-I00 and Unidad de Excelencia María de Maetzu MDM-2016-0692), Xunta de Galicia (Centro singular de investigación de Galicia accreditation 2019-2022, and project ED431C-2021/14), and by the European Union FEDER.

# A    Conventions

In this appendix, we collect facts regarding solutions of the classical Yang-Baxter equation and our related conventions. Let us consider a generic Lie superalgebra $\mathfrak{g}$ (the truncation of these facts to bosonic Lie subalgebras is straightforward). We are after "antisymmetric" constant solutions of the classical Yang-Baxter equation on $\mathfrak{g}$. That means that we want to construct a linear operator

$$R: \ \mathfrak{g} \to \mathfrak{g}, \tag{A.1}$$

that is antisymmetric with respect to a bilinear form on $\mathfrak{g}$. For simplicity here we assume that the latter is implemented by taking the (super)trace in a matrix realisation of $\mathfrak{g}$, so that antisymmetry reads like

$$\mathrm{STr}(Rx \ y) = -\mathrm{STr}(x \ Ry), \qquad \forall x, y \in \mathfrak{g}. \tag{A.2}$$

Moreover, we demand that it solves the classical Yang-Baxter equation which reads

$$[[Rx, Ry]] - R([[Rx, y]] + [[x, Ry]]) = 0, \qquad \forall x, y \in \mathfrak{g}, \tag{A.3}$$

where $[[,]]$ denotes the graded commutator on the superalgebra (i.e. it is the anticommutator when the two elements are odd and the commutator otherwise).

Let us introduce a basis $\mathsf{T}_I$ for $\mathfrak{g}$ to identify the structure constants as $[[T_I, T_J]] = f_{IJ}{}^K T_K$. The linear operator $R$ can be thought of as a matrix $R_J{}^J$ because $R\mathsf{T}_I = R_I{}^J \mathsf{T}_J$. Moreover, after denoting by $K_{IJ} = \mathrm{STr}(\mathsf{T}_I \mathsf{T}_J)$ the metric on $\mathfrak{g}$ and $K^{IJ}$ its inverse, antisymmetry just corresponds to the statement that $R^{IJ} = K^{IK} R_K{}^J$ is antisymmetric in the $I, J$ indices if the indices correspond to even generators of $\mathfrak{g}$, while $R^{IJ}$ is symmetric if the indices correspond to odd generators. This difference is due to an extra sign coming from the supertrace.

We can also map $R$ to an element $r$ of the 2-fold wedge product of $\mathfrak{g}$ by

$$r = -\tfrac{1}{2} R^{IJ} \mathsf{T}_I \wedge \mathsf{T}_J, \tag{A.4}$$

where we use the graded wedge product

$$x \wedge y = x \otimes y - (-1)^{\deg(x) * \deg(y)} y \otimes x. \tag{A.5}$$

In this definition we are using the function deg that gives the degree of the superalgebra element, i.e. it is 0 on even elements and 1 on odd ones. This imples that $\wedge$ is symmetric if both $x$ and $y$ are odd, otherwise it is antisymmetric.

The above algebraic ingredients can be used to construct integrable deformations of 2-dimensional sigma models. In the case of deformations of semisymmetric supercoset sigma-models we assume the existence of a $\mathbb{Z}_4$-grading of $\mathfrak{g}$ such that $\mathfrak{g} = \oplus_{i=0}^3 \mathfrak{g}^{(i)}$ and $[[\mathfrak{g}^{(i)}, \mathfrak{g}^{(j)}]] \subset \mathfrak{g}^{(i+j \bmod 4)}$. Then the action of the deformed sigma model is [7]

$$S = -\frac{\sqrt{\lambda}}{8\pi} \int d\tau d\sigma \, (\sqrt{|h|} h^{mn} - \varepsilon^{mn}) \, \mathrm{STr} \left( J_m \, \hat{d} \frac{1}{1 - \eta R_g \hat{d}} J_n \right). \tag{A.6}$$

We have introduced a worldsheet metric $h_{mn}$ and the antisymmetric tensor $\varepsilon^{\tau\sigma} = -\varepsilon^{\sigma\tau} = -1$. We also have the Maurer-Cartan form $J = g^{-1} dg$ with $g \in G$ and $\mathfrak{g} = \mathrm{Lie}(G)$, and $\hat{d} = \tfrac{1}{2} P^{(1)} + P^{(2)} - \tfrac{1}{2} P^{(3)}$ with $P^{(i)}$ the projectors on $\mathfrak{g}^{(i)}$. The shorthand notation $R_g$ means $R_g = \mathrm{Ad}_g^{-1} R \, \mathrm{Ad}_g$ where $\mathrm{Ad}_g x = gxg^{-1}$ and it is multiplied by the deformation parameter $\eta \in \mathbb{R}$.

It is now very simple to obtain the truncation to symmetric bosonic cosets. In that case, one assumes a $\mathbb{Z}_2$-grading and replaces the supertrace by the trace as well as $\hat{d} = P^{(2)}$. To reduce even further to deformations of the Principal Chiral Model, we only have to take $\hat{d} = 1$.

Importantly, the previous algebraic ingredients can be used also to generate deformations of supergravity backgrounds that do not necessarily correspond to integrable sigma-models. In order to make sure that the deformation still solves the type II supergravity equations, the $R$-matrix must satisfy an additional linear constraint called the "unimodularity condition" [15]

$$0 = K^{IJ}[[\mathsf{T}_I, R\mathsf{T}_J]] = R^{IJ} f_{IJ}{}^K \mathsf{T}_K. \tag{A.7}$$

To conclude, let us remark that we are interested only in real deformations of the supergravity background. Demanding the reality of the sigma-model action we find that the $R$-matrix gives rise to a real deformation if $R^{IJ}$ is anti-hermitian $(R^{JI})^* = -R^{IJ}$. This result follows from assuming the reality condition $T_I^\dagger + H T_I H^{-1} = 0$ for elements of $\mathfrak{g}$. See appendix B for the case of $\mathfrak{psu}(2,2|4)$. As remarked above, on the one hand, $R^{IJ}$ is antisymmetric if the indices correspond to even generators of $\mathfrak{g}$ and then the entries of $R^{IJ}$ must be real. On the other hand, $R^{IJ}$ is symmetric if the indices correspond to odd generators of $\mathfrak{g}$ and then the entries of $R^{IJ}$ must be imaginary.

# B    The $\mathfrak{psu}(2,2|4)$ superalgebra

Here we collect our conventions on the $\mathcal{N} = 4$ superconformal algebra, and we provide an explicit matrix realisation of $\mathfrak{su}(2,2|4)$ that is useful for explicit calculations. Useful reviews are for example [45] and [46].

### Indices conventions

We will use $\mu, \nu = 0, \ldots, 3$ for indices in the 4-dimensional spacetime and we will take the Minkowski metric to be $\eta_{\mu\nu} = \mathrm{diag}(-1, +1, +1, +1,)$. Knowing that the Lorentz algebra can be rewritten as $\mathfrak{so}(1,3) \sim \mathfrak{sl}(2,\mathbb{R})_L \oplus \mathfrak{sl}(2,\mathbb{R})_R$, we will use $\alpha, \beta = 1, 2$ for spinor indices of $\mathfrak{sl}(2,\mathbb{R})_L$ and $\dot\alpha, \dot\beta = 1, 2$ for spinor indices of $\mathfrak{sl}(2,\mathbb{R})_R$. Finally, we will use $a, b = 1, \ldots, 4$ for spinor indices of $SO(6)$, and $A, B = 1, \ldots 6$ for fundamental indices of $SO(6)$.

### The conformal algebra $\mathfrak{so}(2,4)$

The Lorentz algebra is spanned by $J_{\mu\nu}$ satisfying

$$[J_{\mu\nu}, J_{\rho\sigma}] = \eta_{\mu\rho} J_{\nu\sigma} - \eta_{\nu\rho} J_{\mu\sigma} + \eta_{\nu\sigma} J_{\mu\rho} - \eta_{\mu\sigma} J_{\nu\rho}. \tag{B.1}$$

With the addition of $p_\mu$ they form the Poincaré algebra

$$[J_{\mu\nu}, p_\rho] = \eta_{\mu\rho} p_\nu - \eta_{\nu\rho} p_\mu. \tag{B.2}$$

Adding the dilatation generator $D$ and the special conformal generators $k_\mu$ we obtain the full conformal algebra, whose remaining commutation relations are

$$[D, p_\mu] = p_\mu, \qquad [D, k_\mu] = -k_\mu, \qquad [p_\mu, k_\nu] = -2\eta_{\mu\nu} D + 2J_{\mu\nu}, \qquad [J_{\mu\nu}, k_\rho] = \eta_{\mu\rho} k_\nu - \eta_{\nu\rho} k_\mu. \tag{B.3}$$

All other commutation relations are trivial

$$[p_\mu, p_\nu] = [k_\mu, k_\nu] = [D, J_{\mu\nu}] = 0. \tag{B.4}$$

## Supercharges and $R$-symmetry

At this point we introduce supercharges $Q_{\alpha a}, \overline{Q}^{\dot\alpha a}, S_\alpha{}^a, \overline{S}^{\dot\alpha}{}_a$. The generators $J_{\mu\nu}, p_\mu, Q_{\alpha a}, \overline{Q}^{\dot\alpha a}$ span the ($\mathcal{N} = 4$) super-Poincaré algebra. A lower (resp. upper) index $\alpha$ means that they transform in the $\mathbf{2}$ (resp. $\overline{\mathbf{2}}$) representation of $\mathfrak{sl}(2,\mathbb{R})_L$, and similarly for dotted indices of $\mathfrak{sl}(2,\mathbb{R})_R$. To be more explicit, let us define the antisymmetric tensor $\epsilon_{12} = -\epsilon_{21} = -\epsilon^{12} = \epsilon^{21} = 1$ such that $\epsilon^{\alpha\gamma}\epsilon_{\gamma\beta} = \delta^\alpha_\beta$. This is used to raise and lower indices as $\psi^\alpha = \epsilon^{\alpha\beta}\psi_\beta, \psi_\alpha = \epsilon_{\alpha\beta}\psi^\beta$. We then define $(\sigma^\mu)_{\alpha\dot\alpha}$ and $(\overline{\sigma}^\mu)^{\dot\alpha\alpha}$ as

$$\sigma^\mu = -i\,(\mathbf{1}, \sigma^j), \qquad \overline{\sigma}^\mu = -i\,(\mathbf{1}, -\sigma^j), \tag{B.5}$$

where $\sigma^j$ are the Pauli matrices. The matrices $\sigma^\mu, \overline{\sigma}^\mu$ appear in the Weyl representation of the 4-dimensional gamma-matrices

$$\gamma^\mu = \begin{pmatrix} 0 & (\sigma^\mu)_{\alpha\dot\alpha} \\ (\overline{\sigma}^\mu)^{\dot\alpha\alpha} & 0 \end{pmatrix}, \tag{B.6}$$

that satisfy $\{\gamma_\mu, \gamma_\nu\} = 2\eta_{\mu\nu}\mathbf{1}$. At this point we define

$$(\sigma^{\mu\nu})_\alpha{}^\beta = \tfrac{1}{4}(\sigma^\mu\overline{\sigma}^\nu - \sigma^\nu\overline{\sigma}^\mu)_\alpha{}^\beta, \qquad (\overline{\sigma}^{\mu\nu})^{\dot\alpha}{}_{\dot\beta} = \tfrac{1}{4}(\overline{\sigma}^\mu\sigma^\nu - \overline{\sigma}^\nu\sigma^\mu)^{\dot\alpha}{}_{\dot\beta}, \tag{B.7}$$

so that we can write the commutators of the supercharges with the Lorentz generators

$$\begin{aligned} [J_{\mu\nu}, Q_{\alpha a}] &= (\sigma_{\mu\nu})_\alpha{}^\beta Q_{\beta a}, & [J_{\mu\nu}, \overline{Q}^{\dot\alpha a}] &= (\overline{\sigma}_{\mu\nu})^{\dot\alpha}{}_{\dot\beta} \overline{Q}^{\dot\beta a}, \\ [J_{\mu\nu}, S_\alpha{}^a] &= (\sigma_{\mu\nu})_\alpha{}^\beta S_\beta{}^a, & [J_{\mu\nu}, \overline{S}^{\dot\alpha}{}_a] &= (\overline{\sigma}_{\mu\nu})^{\dot\alpha}{}_{\dot\beta} \overline{S}^{\dot\beta}{}_a. \end{aligned} \tag{B.8}$$

We have the following trivial commutation relations

$$[p_\mu, Q_{\alpha a}] = [p_\mu, \overline{Q}^{\dot\alpha a}] = [k_\mu, S_\alpha{}^a] = [k_\mu, \overline{S}^{\dot\alpha}{}_a] = 0, \tag{B.9}$$

and the commutation relations with the dilatation generator

$$\begin{aligned} [D, Q_{\alpha a}] &= \tfrac{1}{2}Q_{\alpha a}, & [D, S_\alpha{}^a] &= -\tfrac{1}{2}S_\alpha{}^a, \\ [D, \overline{Q}^{\dot\alpha a}] &= \tfrac{1}{2}\overline{Q}^{\dot\alpha a}, & [D, \overline{S}^{\dot\alpha}{}_a] &= -\tfrac{1}{2}\overline{S}^{\dot\alpha}{}_a. \end{aligned} \tag{B.10}$$

The commutators relating the $Q$ and $S$ supercharges are

$$\begin{aligned} [k^\mu, Q_{\alpha a}] &= +i\sigma^\mu_{\alpha\dot\alpha} \overline{S}^{\dot\alpha}{}_a, & [k_\mu, \overline{Q}^{\dot\alpha a}] &= -i\overline{\sigma}_\mu^{\dot\alpha\alpha} S_\alpha{}^a, \\ [p^\mu, S_\alpha{}^a] &= -i\sigma^\mu_{\alpha\dot\alpha} \overline{Q}^{\dot\alpha a}, & [p_\mu, \overline{S}^{\dot\alpha}{}_a] &= +i\overline{\sigma}_\mu^{\dot\alpha\alpha} Q_{\alpha a}. \end{aligned} \tag{B.11}$$

The $\mathcal{N} = 4$ superconformal algebra has an $SU(4) \sim SO(6)$ $\mathcal{R}$-symmetry, under which the supercharges transform in the $\mathbf{4}$ or $\overline{\mathbf{4}}$ representations (respectively for upper or lower indices $a, b, = 1, \ldots, 4$). We denote the $\mathcal{R}$-symmetry generators as $\mathcal{R}_{AB}$ with $\mathcal{R}_{AB} = -\mathcal{R}_{BA}$ and $A, B = 1, \ldots, 6$. They satisfy the commutation relations

$$[\mathcal{R}_{AB}, \mathcal{R}_{CD}] = \delta_{AC}\mathcal{R}_{BD} - \delta_{BC}\mathcal{R}_{AD} + \delta_{BD}\mathcal{R}_{AC} - \delta_{AD}\mathcal{R}_{BC}, \tag{B.12}$$

and they commute with all generators of the conformal algebra. The action of the $\mathcal{R}$-symmetry generators on the supercharges yields

$$\begin{aligned} [\mathcal{R}_{AB}, Q_{\alpha a}] &= \tfrac{1}{2}(\rho_{AB})_a{}^b Q_{\alpha b}, & [\mathcal{R}_{AB}, \overline{Q}^{\dot\alpha a}] &= -\tfrac{1}{2}(\rho_{AB})_b{}^a \overline{Q}^{\dot\alpha b} \\ [\mathcal{R}_{AB}, \overline{S}^{\dot\alpha}{}_a] &= \tfrac{1}{2}(\rho_{AB})_a{}^b \overline{S}^{\dot\alpha}{}_b, & [\mathcal{R}_{AB}, S_\alpha{}^a] &= -\tfrac{1}{2}(\rho_{AB})_b{}^a S_\alpha{}^b. \end{aligned} \tag{B.13}$$

Indices $A, B$ will be raised and lowered with the Kronecker delta. Simple anticommutators are

$$\{Q_{\alpha a}, \overline{Q}_{\dot{\alpha}}{}^b\} = \delta_a^b \, \sigma_{\alpha\dot{\alpha}}^\mu \, p_\mu, \qquad\qquad \{S_\alpha{}^a, \overline{S}_{\dot{\alpha} b}\} = -\delta_b^a \, \sigma_{\alpha\dot{\alpha}}^\mu \, k_\mu. \qquad (B.14)$$

The trivial mixed anticommutators are

$$\{Q_{\alpha a}, \overline{S}^{\dot{\beta}}{}_b\} = 0, \qquad \{\overline{Q}^{\dot{\alpha} a}, S_\beta{}^b\} = 0, \qquad (B.15)$$

while the remaining non-trivial mixed anticommutators

$$\{Q_{\alpha a}, S_\beta{}^b\} = \tfrac{i}{2} \, \epsilon_{\alpha\beta} \, (\rho^{AB})_a{}^b \, \mathcal{R}_{AB} + i \, \delta_a^b \, \sigma_{\alpha\beta}^{\mu\nu} \, J_{\mu\nu} + i \, \epsilon_{\alpha\beta} \, \delta_a^b \, D + \tfrac{i}{2} \, \epsilon_{\alpha\beta} \, \delta_a^b \, \mathbf{1},$$
$$\{\overline{Q}^{\dot{\alpha} a}, \overline{S}^{\dot{\beta}}{}_b\} = \tfrac{i}{2} \, \epsilon^{\dot{\alpha}\dot{\beta}} \, (\rho^{AB})_b{}^a \, \mathcal{R}_{AB} - i \, \delta_b^a \, \overline{\sigma}_{\mu\nu}^{\dot{\alpha}\dot{\beta}} \, J^{\mu\nu} - i \, \epsilon^{\dot{\alpha}\dot{\beta}} \, \delta_b^a \, D + \tfrac{i}{2} \, \epsilon^{\dot{\alpha}\dot{\beta}} \, \delta_b^a \, \mathbf{1}. \qquad (B.16)$$

The relations that we are writing here actually correspond to the $\mathfrak{su}(2,2|4)$ superalgebra. To obtain the relations of $\mathfrak{psu}(2,2|4)$ (which is isomorphic to the $\mathcal{N} = 4$ superconformal algebra) one has to project out the identity operator.

## Matrix realisation

In the anticommutators above we included the terms proportional to the identity operator because we want to give an explicit matrix realisation of the superalgebra, and $\mathfrak{su}(2,2|4)$ admits one while $\mathfrak{psu}(2,2|4)$ does not. To obtain the matrix realisation we start from the above definition (B.6) of the gamma matrices, which is also equivalent to

$$\gamma^0 = -i\sigma^1 \otimes \mathbf{1}_2, \qquad \gamma^1 = \sigma^2 \otimes \sigma^1, \qquad \gamma^2 = \sigma^2 \otimes \sigma^2, \qquad \gamma^3 = \sigma^2 \otimes \sigma^3, \qquad (B.17)$$

and we supplement them with

$$\gamma^4 = -\sigma^3 \otimes \mathbf{1}_2, \qquad (B.18)$$

to obtain gamma matrices in 5 dimensions. In general, we define $\gamma_{mn} = \frac{1}{2}[\gamma_m, \gamma_n]$ for any gamma. We also define

$$\tilde{\gamma}_i = \gamma_i, \quad i = 1, \ldots, 4, \qquad \tilde{\gamma}_5 = i \, \gamma_0, \qquad (B.19)$$

which are gamma-matrices in 5 *Euclidean* dimensions. We use them to define the matrices $\rho_{AB}$ which are antisymmetric in the indices $A, B = 1, \ldots, 6$ as

$$\rho_{AB} = \tilde{\gamma}_{AB}, \quad A, B = 1, \ldots, 5, \qquad \rho_{A6} = -i \, \tilde{\gamma}_A. \qquad (B.20)$$

Finally, we have everything we need to construct a matrix realisation of $\mathfrak{su}(2,2|4)$ in terms of $8 \times 8$ matrices. For the generators of the conformal algebra we take

$$J_{\mu\nu} = \begin{pmatrix} -\frac{1}{2}\gamma_{\mu\nu} & \mathbf{0}_4 \\ \mathbf{0}_4 & \mathbf{0}_4 \end{pmatrix}, \qquad p_\mu = \begin{pmatrix} -\frac{1}{2}(\gamma_{\mu 4} + \gamma_\mu) & \mathbf{0}_4 \\ \mathbf{0}_4 & \mathbf{0}_4 \end{pmatrix},$$
$$D = \begin{pmatrix} -\frac{1}{2}\gamma_4 & \mathbf{0}_4 \\ \mathbf{0}_4 & \mathbf{0}_4 \end{pmatrix}, \qquad k_\mu = \begin{pmatrix} -\frac{1}{2}(\gamma_{\mu 4} - \gamma_\mu) & \mathbf{0}_4 \\ \mathbf{0}_4 & \mathbf{0}_4 \end{pmatrix}. \qquad (B.21)$$

Similarly, for the $\mathcal{R}$-symmetry generators

$$\mathcal{R}_{AB} = \begin{pmatrix} \mathbf{0}_4 & \mathbf{0}_4 \\ \mathbf{0}_4 & -\frac{1}{2}\rho_{AB} \end{pmatrix}. \qquad (B.22)$$

To conclude, the supercharges are realised as

$$Q^\alpha{}_a = \sqrt{2} \begin{pmatrix} \mathbf{0}_4 & E_{\alpha, a} \\ \mathbf{0}_4 & \mathbf{0}_4 \end{pmatrix}, \qquad \overline{Q}^{\dot{\alpha} a} = -\sqrt{2} \begin{pmatrix} \mathbf{0}_4 & \mathbf{0}_4 \\ E_{a, \dot{\alpha}+2} & \mathbf{0}_4 \end{pmatrix},$$
$$S_\alpha{}^a = i\sqrt{2} \begin{pmatrix} \mathbf{0}_4 & \mathbf{0}_4 \\ E_{a, \alpha} & \mathbf{0}_4 \end{pmatrix}, \qquad \overline{S}_{\dot{\alpha} a} = i\sqrt{2} \begin{pmatrix} \mathbf{0}_4 & E_{\dot{\alpha}+2, a} \\ \mathbf{0}_4 & \mathbf{0}_4 \end{pmatrix}, \qquad (B.23)$$

where $E_{a,b}$ are the $4 \times 4$ unit matrices with zeros everywhere, except 1 at position $a, b$.

## Reality condition on the superalgebra

To write down the reality condition, let us define

$$H = \begin{pmatrix} -i\gamma_0 & \mathbf{0}_4 \\ \mathbf{0}_4 & \mathbf{1}_4 \end{pmatrix}. \tag{B.24}$$

With this choice $H^\dagger = H$, where $\dagger$ denotes conjugate-transpose. For all bosonic generators $X$ (i.e. from the conformal or the $\mathcal{R}$-symmetry algebra) the reality condition is satisfied as $X^\dagger + HXH^{-1} = 0$. For supercharges, instead, the dagger relates the barred and unbarred supercharges in the following way

$$(Q_{\alpha a})^\dagger + H\overline{Q}_\alpha{}^a H^{-1} = 0, \qquad (S_\alpha{}^a)^\dagger + H\overline{S}_{\alpha a} H^{-1} = 0. \tag{B.25}$$

This means that we are using a complex basis. A generic element $M$ of $\mathfrak{psu}(2,2|4)$, however, is required to satisfy simply $M^\dagger + HMH^{-1} = 0$.

## $\mathbb{Z}_4$ automorphism

The $\mathcal{N} = 4$ superconformal algebra admits a $\mathbb{Z}_4$ automorphism. Let us define the matrix

$$K_{ab} = -i(\mathbf{1}_2 \otimes \sigma_2)_{ab}, \tag{B.26}$$

and let us denote by $K^{ab}$ its inverse, so that $K^{ac}K_{cb} = \delta_b^a$. We will use $K$ to raise and lower $a, b$ indices, with the same conventions as for the Lorentz indices, namely $V_a = K_{ab}V^b$, $V^a = K^{ab}V_b$. We also define

$$\mathcal{K} = \mathbf{1}_2 \otimes K, \tag{B.27}$$

which we use for the definition of the $\mathbb{Z}_4$ automorphism as

$$\Omega(X) = -\mathcal{K} X^{st} \mathcal{K}^{-1}, \tag{B.28}$$

where $st$ denotes supertransposition. The $\mathbb{Z}_4$ automorphism induces the decomposition $\mathfrak{g} = \oplus_{i=0}^3 \mathfrak{g}^{(i)}$ of the superalgebra and one can construct the projectors $P^{(i)}$ on each of these subspaces. Then we find the following decomposition

$$\begin{aligned}
p_\mu + k_\mu, \ J_{\mu\nu}, \ \mathcal{R}_{\bar{A}\bar{B}} &\in \mathfrak{g}^{(0)} & Q_{\alpha a} + S_{\alpha a}, \ \overline{Q}^{\dot\alpha a} - \overline{S}^{\dot\alpha a} &\in \mathfrak{g}^{(1)}, \\
p_\mu - k_\mu, \ D, \ \mathcal{R}_{\bar{A}6} &\in \mathfrak{g}^{(2)}, & Q_{\alpha a} - S_{\alpha a}, \ \overline{Q}^{\dot\alpha a} + \overline{S}^{\dot\alpha a} &\in \mathfrak{g}^{(3)},
\end{aligned} \tag{B.29}$$

where above $\bar{A}, \bar{B} = 1, \dots, 5$. We remind that indices are raised and lowered with $\epsilon$ and $K$. As reviewed above, in the construction of the supercoset action and its deformations, one introduces a particular combination of the projectors which is $\hat{d} = P^{(1)} + 2P^{(2)} - P^{(3)}$. The action of its transpose $\hat{d}^T = -P^{(1)} + 2P^{(2)} + P^{(3)}$ on the supercharges is

$$\hat{d}^T(Q_{\alpha a}) = -S_{\alpha a}, \qquad \hat{d}^T(S_{\alpha a}) = -Q_{\alpha a}, \qquad \hat{d}^T(\overline{Q}^{\dot\alpha a}) = \overline{S}^{\dot\alpha a}, \qquad \hat{d}^T(\overline{S}^{\dot\alpha a}) = \overline{Q}^{\dot\alpha a}. \tag{B.30}$$

## Supertrace relations

The non-vanishing relations involving the supertrace are

$$\begin{aligned}
&\mathrm{STr}(p_\mu k_\nu) = 2\eta_{\mu\nu}, & &\mathrm{STr}(J_{\mu\nu}J_{\rho\sigma}) = -(\eta_{\mu\rho}\eta_{\nu\sigma} - \eta_{\nu\rho}\eta_{\mu\sigma}), \\
&\mathrm{STr}(DD) = 1, & &\mathrm{STr}(\mathcal{R}_{AB}\mathcal{R}_{CD}) = \delta_{AC}\delta_{BD} - \delta_{BC}\delta_{AD} \\
&\mathrm{STr}(Q_{\alpha a}S_\beta{}^b) = 2i\,\epsilon_{\alpha\beta}\delta_a^b, & &\mathrm{STr}(\overline{Q}^{\dot\alpha a}\overline{S}^{\dot\beta}{}_b) = -2i\,\epsilon^{\dot\alpha\dot\beta}\delta_b^a.
\end{aligned} \tag{B.31}$$

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
