# Peer review of "All Jordanian deformations of the AdS5 $\times$ S5 superstring"

_SciPost Physics_

## Round 1 · Referee Report · Stijn van Tongeren (Referee 1) · 2023-2-7

Strengths

1 - Systematic approach addressing a relevant problem in the field, determining the possible integrable deformations of integrable superstrings of Jordanian type.
2 - Clear discussion of an inherently technical topic, with an appropriate level of detail.

Weaknesses

1 - The residual symmetry analysis is not carried out for all cases.
2 - For some extended bosonic r matrices the automorphism analysis is subtle and not fully settled.

Report

This paper classifies the possible jordanian Yang-Baxter deformations of the AdS5xS5 superstring. These integrability preserving deformations are highly relevant to new developments in integrability in the AdS/CFT correspondence. Jordanian deformations are algebraically nontrivial, yet have a structure that may make them accessible at the quantum level as well, which would open exciting possibilities in the context of AdS/CFT.

After a helpful summary of results, the paper introduces jordanian deformations and r matrices, and outlines their general approach to classifying them. The classification then proceeds by considering possible embeddings of the nonabelian two dimensional Lie algebra at the basis of jordanian deformations, in so(2,4), up to automorphisms. Next, the possible bosonic extensions are discussed, completing a classification of bosonic jordnaian deformations of the superconformal algebra psu(2,2|4). Then, the authors consider unimodular versions of their jordanian r matrices, motivated by a corresponding constraint arising in the associated deformed string sigma models. Unimodularity requires fermionic (supercharge) extensions, and the authors discuss all such extensions of their bosonic result, providing a wide range of new r matrices that can be used to generate integrable deformations of the AdS5xS5 superstring.

In terms of the expectations listed in the acceptance criteria, the paper opens up a new area of integrable deformations for study. All general acceptance criteria are clearly met, provided the changes below are implemented.

The referee instructions specifically request that I also comment on the suitability of this article for this particular journal. I believe this would come down to publication in SciPost Physics, or Scipost Physics Core. Given the relative youth of both journals, especially the second, it is difficult to make an intuitive judgement here. I presume that some years from now, the majority of submissions should appear in Scipost Physics Core, with Scipost Physics attaining a special status (of currently undetermined type). At present the majority of papers appear to be published in Scipost Physics. Since I would rate the quality of this paper significantly above average, I believe this is the appropriate journal for this submission.

Requested changes

  • The place where reference [27] is placed in "[...] to the case where the subalgebra participating in the construction of R is non-abelian [27]." may give the undesired impression that [27] considered non-abelian cases. It would be good to rephrase this sentence and/or surrounding text.

  • The sentence "In the case of “diagonal” TsT-models, where the object causing the twisting is diagonal, this has been crucial in the understanding of their spectral problem on both sides of deformed AdS5/SYM by means of integrability methods [29–31]" is not quite accurate, since for diagonal deformations of AdS5, no explicit deformation of SYM was known, at least until the recent https://arxiv.org/abs/2301.08757 (with open questions remaining) that appeared after this paper.

  • It would be nice to add some words recalling the main idea of how the possible embeddings of e are identified (bottom page 9), to keep the flow of the presentation.

  • Regarding "these subalgebras" in "because these generators never appear on the right-hand side of the commutation relations of these subalgebras.", I presume this refers to the subalgebras leaving e invariant, rather than s1 and s2 as the sentence might be taken. It would be good to make this explicit (or give a few words why s1 and s2 are actually the relevant algebras to consider in relation to automorphisms).

  • The discussion of removing parameters using automorphisms, is presented in terms of actively removing parameters, but this means it is not as obvious that one cannot do more (in certain cases). If possible, it would be good to add a comment clarifying this.

  • Regarding the extension of bosonic r matrices into so(6), maybe it is nice for the reader to complete the argument explicitly, by pointing out that given the direct sum structure of the algebra, the restriction of the r matrix to so(6) must solve the CYBE over so(6), meaning it can only involve generators in an abelian subalgebra.

  • I would ask the authors to rephrase the second sentence in "Our results may therefore offer a wide range of applications for deformations of the AdS5/SYM holographic duality. See [41–44] for some preliminary proposals of deformations of the dual gauge theory." of the conclusions, taking into account that [43,44] did not present an original proposal, but merely repeated the proposal of [41,42], verbatim. Moreover, the new evidence claimed in [43,44], in favor of the proposal of [41,42], is dubious.

  • typo: "beaing"

  • I find it funny (but not problematic) that the action of the YB sigma model appears in a section just called "Conventions".

The following are more questions than requested changes.

  • Regarding the paragraph "obstructions to the unimodularity extension", this seems reminiscent of the discussion regarding extensions of d^p in [26]. Admittedly this was a simpler setting, but the argument seemed simpler (less technical) as well. Can similar simple statements not cover this here?

  • While not doing a complete analysis, it is possible to comment on the range of possible supersymmetry in the extended cases (e.g. 0-8, 0-12?)

  • validity: top
  • significance: high
  • originality: good
  • clarity: high
  • formatting: excellent
  • grammar: excellent

Author:  Riccardo Borsato  on 2023-03-01  [id 3416]

(in reply to Report 1 by Stijn van Tongeren on 2023-02-07)

We thank the referee for the detailed comments and suggestions. In the following we address each point repeating the comment of the referee.

Ref. (1): The place where reference [27] is placed in "[...] to the case where the subalgebra participating in the construction of R is non-abelian [27]." may give the undesired impression that [27] considered non-abelian cases. It would be good to rephrase this sentence and/or surrounding text.

We now cite [27] in the new sentence "See [27] for the reformulation of TsT transformations as Yang-Baxter deformations."

Ref. (2): The sentence "In the case of “diagonal” TsT-models, where the object causing the twisting is diagonal, this has been crucial in the understanding of their spectral problem on both sides of deformed AdS5/SYM by means of integrability methods [29–31]" is not quite accurate, since for diagonal deformations of AdS5, no explicit deformation of SYM was known, at least until the recent https://arxiv.org/abs/2301.08757 (with open questions remaining) that appeared after this paper.

We have reformulated the text as follows, and have added the new reference [NCYM = https://arxiv.org/abs/2301.08757]:
"In the case of “diagonal” TsT-models, where the object causing the twisting is diagonal, this is crucial in the understanding of their spectral problem by means of integrability methods [29–31].\footnote{From the point of view of the holographic duality, understanding these deformations as deformations of the super Yang-Mills action is not always obvious, in particular when on the string side of the correspondence the AdS space participates in the deformation. We refer to [NCYM] for recent developments in this direction.}"

Ref. (3): It would be nice to add some words recalling the main idea of how the possible embeddings of e are identified (bottom page 9), to keep the flow of the presentation.

The idea to find the possible embeddings of e is essentially the same we used to find the ones for h. One starts from the most generic ansatz and then acts with inner SO(2,4) automorphisms to remove as many free parameters as possible. When doing the computation one sees that different "branches" are possible in the simplification, and this gives rise to the possible *inequivalent* embeddings. Unless the referee thinks that this is not enough, to avoid repetitions we have therefore simply added a footnote around the beginning of the next page after "... inequivalent embeddings of h.", saying
"\footnote{This is essentially the idea that was used in [15] to identify the possible embeddings of e. Notice that in the case of e one does not include in the initial ansatz the generators that cannot appear on the right-hand-side of the commutation relations (i.e. J_{03}, J_{12} and D in the case of s1, and J_{12}, J_{03}-D and k_0+k_3+2p_3 in the case of s_2).}"

Ref. (4): Regarding "these subalgebras" in "because these generators never appear on the right-hand side of the commutation relations of these subalgebras.", I presume this refers to the subalgebras leaving e invariant, rather than s1 and s2 as the sentence might be taken. It would be good to make this explicit (or give a few words why s1 and s2 are actually the relevant algebras to consider in relation to automorphisms).

In fact by "these subalgebras" we mean s1 and s2. The point is that if an inner SO(2,4) automorphism that simplifies h exists, there must be a Lie-algebra element x such that [x,e]=0 (or [x,e] proportional to e, this would not change the argument) and ad_x^n(h) belongs to s_i. One concludes that the addition of x to h and e gives rise (possibly after adding other elements to make it close into an algebra) to a Lie algebra that is again solvable. Therefore, it must be a subalgebra of either s1 or s2, which are maximal solvable. In other words, when looking for inner SO(2,4) automorphisms, one can restrict oneself to inner automorphisms of s1 or s2 only. In any case, it is also possible to check that, after using inner automorphisms of s1 or s2, there is no other inner SO(2,4) automorphism that is useful in removing the generators mentioned in the text. We have added a comment in the text after (4.1) pointing out that the only relevant inner automorphisms are those generated by s1 or s2.

Ref. (5): The discussion of removing parameters using automorphisms, is presented in terms of actively removing parameters, but this means it is not as obvious that one cannot do more (in certain cases). If possible, it would be good to add a comment clarifying this.

We do not understand what the referee has in mind, and what else could be done. Any suggestion in this direction would be very much appreciated. A comment that may be relevant for this point is that sometimes the action of automorphisms on the parameter space can be quite complicated. For example, there may be also some discrete automorphisms (for example \xi --> -\xi and e_{\pm a} --> \pm e_{\pm a}) that effectively constrain the region in parameter space corresponding to inequivalent solutions (see for example R_{14} in (5.10)).

Ref. (6): Regarding the extension of bosonic r matrices into so(6), maybe it is nice for the reader to complete the argument explicitly, by pointing out that given the direct sum structure of the algebra, the restriction of the r matrix to so(6) must solve the CYBE over so(6), meaning it can only involve generators in an abelian subalgebra.

Not all of the so(6) shifts that we had mentioned in the previous comment were possible, actually. We have now updated the discussion on Jordanian solutions in so(2,4)+so(6). Now there is also a new section 6.3 discussing the unimodular extensions in the presence of the so(6) shifts.

Ref. (7): I would ask the authors to rephrase the second sentence in "Our results may therefore offer a wide range of applications for deformations of the AdS5/SYM holographic duality. See [41–44] for some preliminary proposals of deformations of the dual gauge theory." of the conclusions, taking into account that [43,44] did not present an original proposal, but merely repeated the proposal of [41,42], verbatim. Moreover, the new evidence claimed in [43,44], in favor of the proposal of [41,42], is dubious.

We have rephrased it and we have written "See [41–42] for some preliminary proposals of deformations of the dual gauge theory. See also [43-44] for later works."

Ref. (8): typo: "beaing"

Thank you.

Ref. (9): I find it funny (but not problematic) that the action of the YB sigma model appears in a section just called "Conventions".

We have renamed that appendix to "Homogeneous Yang-Baxter deformations."

Ref. (10): The following are more questions than requested changes.
Regarding the paragraph "obstructions to the unimodularity extension", this seems reminiscent of the discussion regarding extensions of d^p in [26]. Admittedly this was a simpler setting, but the argument seemed simpler (less technical) as well. Can similar simple statements not cover this here?

Yes, the calculations that we do are essentially equivalent to the logic used in [26] for d/\p, with p spacelike, and we have pointed it out in the text. We did the calculation in the way presented in the text because we wanted to have a systematic way to look at each possible case, and then in the text we presented the explicit calculations for just one example.

Ref. (10): While not doing a complete analysis, it is possible to comment on the range of possible supersymmetry in the extended cases (e.g. 0-8, 0-12?)

When it comes to the extended cases, we have only checked explicitly that \bar{R}_{12} does not preserve any superisometry. The other cases are complicated by the presence of a large number of possible parameters in the r matrices. It is natural to expect that the unimodular versions of the extended r matrices will preserve less superisometries compared to the unimodular versions of the rank-2 r-matrices from which they are constructed. This is related to the fact that the bosonic extensions are breaking at least some of the residual bosonic isometries present before the extension. It is natural to expect that for a generic choice of the parameters the superisometries will be completely broken, and we added a small comment in the text about that. In the conclusions we refer to this question as an outlook.

---

## Round 1 · Referee Report · Anonymous (Referee 2) · 2023-2-8

Strengths

1- The paper provides a clear and structured classification of Jordanian deformations of the AdS5 superstring. 2- The paper gives a good presentation of their approach, conventions and technical details while staying readable.

Weaknesses

1-They only studied the isometries related to the unimodular extension for the rank 2 case.

Report

In this paper the authors present a full classification of Jordanian deformations of the AdS5xS5 superstring. These deformations are classified by solutions of the (modified) classical Yang-Baxter equation that are of Jordanian type. They then carry out their classification working through the different possible ranks and solutions. They also discuss whether their solutions admit fermionic extensions such that it becomes unimodular. The classification results are listed in section 2 and the remainder of the paper deals with the technical details.

The paper is well written and clear. It is also of interest to the community. There are several new classical r-matrices and corresponding deformations whose corresponding deformations can now be studied. For this reason I recommend this paper for publication.

Requested changes

1- I noticed several typos, such as "siginificant", "beaing" and it would be good to double check the paper for other misspellings that have been missed.

  • validity: high
  • significance: high
  • originality: good
  • clarity: high
  • formatting: good
  • grammar: excellent

Author:  Riccardo Borsato  on 2023-03-01  [id 3417]

(in reply to Report 2 on 2023-02-08)

We thank the referee for the comments and for spotting the typos. We have revised the text to check also other possible typos.

---

## Round 1 · Referee Report · Anonymous (Referee 3) · 2023-2-9

Report

In the paper under review the authors classify bosonic jordanian solutions of the classical Yang-Baxter equation for the Lie algebra $\mathfrak{so}(2,4)$, finding solutions of rank 2, 4 and 6. They then proceed to classify unimodular extensions of these r-matrices for the Lie superalgebra $\mathfrak{psu}(2,2|4)$. The classifications are mostly done up to inner automorphisms. Finally, a symmetry analysis of the unimodular extensions of rank 2 r-matrices is carried out, including a count of the number of commuting supercharges. This determines the symmetries of the corresponding deformed string sigma model, or equivalently the superisometries of the deformed supergravity background.

The results contribute to the more general goal of classifying and understanding integrable deformations of the $AdS_5 \times S^5$ string sigma model. The classifications are carried out systematically and the paper contains detailed discussions of different branches of solutions and special cases. It is well written and the results are collected and clearly presented in tables. While there may be more elegant approaches to some derivations, the direct approach taken means that the paper is straightforward to follow and the results are reproducible. The Conclusions contain a nice summary of the paper, but there is room to expand on possible generalisations, open questions and links with other work.

Having read the paper, I only have a few minor comments and questions, which I have listed below. The paper is of a high quality, contains original results and is clearly of interest in the study of integrable deformations. I recommend the paper for publication in SciPost Physics once the authors have had an opportunity to consider the comments below.

  1. On page 3, the second paragraph of the summary of rank 2 r-matrices seems out of place. The shifts that are described lead to r-matrices outside the jordanian class and, as the authors write, a detailed analysis of such shifts goes beyond the scope of the paper. I would suggest that this paragraph would make more sense in the Conclusions motivated by the broader goal of classifying all possible unimodular r-matrices for $\mathfrak{psu}(2,2|4)$.

  2. On page 9, it would be helpful to state what values the index $i$ takes in $\mathfrak{s}_1$ in eq. (4.1).

  3. On page 10, the authors highlight the existence of inner automorphisms that simply rescale the r-matrix. In principle, these have a physical consequence since they allow the "deformation" parameter to be set to a fixed value. It would be interesting to comment on the existence of such inner automorphisms for unimodular r-matrices.

  4. On page 26, the authors explain that they do not carry out a symmetry analysis of higher-rank unimodular jordanian r-matrices due to the large number of parameters. Is there a general expectation for the number of supersymmetries in these cases, i.e. for a generic choice of parameters?

  5. The authors use $\mathfrak{so}(2,4)$ and $\mathfrak{so}(6)$ inner automorphisms to relate equivalent r-matrices. While this certainly means that the two sigma models are formally related by a field redefinition, it seems plausible that there may be subtleties relating the two quantum theories, e.g. the identification of charges, and thus deformations of the AdS$_5$/CFT$_4$ duality. Given the classification is carried out up to inner automorphisms, it would be interesting to hear any comments the authors have on this issue.

  6. I also spotted the following typographical errors: i. Page 4, 2 lines beneath eq. (2.2): "on" -> "for" ii. Page 5, 1 line beneath eq. (2.5): "in" -> "by" iii. Page 9, 1st line of foot. 7: "nor" -> "neither" iv. Page 17, 1st line of sec. 6.1: "that" -> "which" v. Page 23, 4th line of foot. 23: "sign to" -> "sign of" vi. Page 25, 1st line of para. 2: "originate the" -> "originate from the"

  • validity: top
  • significance: high
  • originality: high
  • clarity: high
  • formatting: excellent
  • grammar: excellent

Author:  Riccardo Borsato  on 2023-03-01  [id 3418]

(in reply to Report 3 on 2023-02-09)

We thank the referee for the detailed comments and suggestions.

  1. Not all of those shifts were possible, actually, and we have updated the discussion on Jordanian solutions in so(2,4)+so(6). The current class of so(6)-shifts do, in fact, fall in the Jordanian class. Now there is also a new section 6.3 discussing the unimodular extensions in the presence of the so(6) shifts and in the summary section the old paragraph got a designated place at the end of it.

  2. We changed it and wrote the full list p0,p1,p2,p3 instead. In the text afterwards we have also spotted various uses of the index "i" which we changed for "\mu" (=0,1,2,3), which was the appropriate index introduced in appendix B.

  3. Using (3.3) it is easy to see that an inner automorphism generated by h has the effect of producing an overall rescaling for the generic extended r matrix (unimodular or non-unimodular) in (3.4). We have added a small paragraph in the conclusions to point this out, and to remind the reader of the non-abelian T-duality interpretation of this fact.

  4. When it comes to the extended cases, we have only checked explicitly that \bar{R}_{12} does not preserve any superisometry. The other cases are complicated by the presence of a large number of possible parameters in the r matrices. It is natural to expect that the unimodular versions of the extended r matrices will preserve less superisometries compared to the unimodular versions of the rank-2 r-matrices from which they are constructed. This is related to the fact that the bosonic extensions are breaking at least some of the residual bosonic isometries present before the extension. It is natural to expect that for a generic choice of the parameters the superisometries will be completely broken, and we added a small comment in the text about that. In the conclusions we refer to this question as an outlook.

  5. We do not share the worry of the referee on this point. Perhaps they have in mind subtleties that may be understood for example as the measure in the path integral not being invariant under the field redefinition. But we don't understand why this possibility should be "plausible". Let's consider two deformed sigma models that are related to each other by a local field redefinition as the ones considered in our paper. The deformed models are continuously connected to the undeformed sigma-model on AdS5xS5. If the field redefinition (that is independent of the deformation parameter eta) caused an anomaly, one would expect the same anomaly to survive in the eta --> 0 limit. But this is not possible, because for the undeformed theory the field redefinition that we are considering is actually part of the psu(2,2|4) symmetry.

We hope that we have addressed the point raised by the referee, but we might have misinterpreted the comment, so we look forward to any further comments they may have.

6 . Thank you for spotting them. We have revised the text to check also other possible typos.

Finally, we have also expanded the Conclusions.

---

## Editorial Decision

resubmitted